# Solving a Class of Non-Convex Minimax Optimization in Federated Learning

**Xidong Wu**[†]
Electrical and Computer Engineering
University of Pittsburgh
Pittsburgh, PA 15213
xidong_wu@outlook.com

**Jianhui Sun**[†]
Computer Science
University of Virginia
Charlottesville, VA 22903
js9gu@virginia.edu

**Zhengmian Hu**
Computer Science
University of Maryland
College Park, MD 20742
huzhengmian@gmail.com

**Aidong Zhang**[§]
Computer Science
University of Virginia
Charlottesville, VA 22903
aidong@virginia.edu

**Heng Huang**[*]
Computer Science
University of Maryland
College Park, MD 20742
henghuanghh@gmail.com

## Abstract

The minimax problems arise throughout machine learning applications, ranging from adversarial training and policy evaluation in reinforcement learning to AUROC maximization. To address the large-scale distributed data challenges across multiple clients with communication-efficient distributed training, federated learning (FL) is gaining popularity. Many optimization algorithms for minimax problems have been developed in the centralized setting (*i.e.*, single-machine). Nonetheless, the algorithm for minimax problems under FL is still underexplored. In this paper, we study a class of federated nonconvex minimax optimization problems. We propose FL algorithms (FedSGDA+ and FedSGDA-M) and reduce existing complexity results for the most common minimax problems. For nonconvex-concave problems, we propose FedSGDA+ and reduce the communication complexity to $O(\varepsilon^{-6})$. Under nonconvex-strongly-concave and nonconvex-PL minimax settings, we prove that FedSGDA-M has the best-known sample complexity of $O(\kappa^3 N^{-1}\varepsilon^{-3})$ and the best-known communication complexity of $O(\kappa^2\varepsilon^{-2})$. FedSGDA-M is the first algorithm to match the best sample complexity $O(\varepsilon^{-3})$ achieved by the single-machine method under the nonconvex-strongly-concave setting. Extensive experimental results on fair classification and AUROC maximization show the efficiency of our algorithms.

## 1 Introduction

The nonconvex minimax optimization has been actively applied to solve enormous machine learning problems, such as adversarial training [56, 61], generative adversarial networks (GANs) [16, 18], policy evaluation in reinforcement learning [55, 22, 25], robust optimization [11, 60], AUROC (area under the ROC curve) maximization [39], *etc*. Many single-machine minimax optimization algorithms have been proposed to address these problems.

---

[†]Equal contribution

[§]This work was partially supported by NSF CNS 2213700 and CCF 2217071 at UVA.

[*]This work was partially supported by NSF IIS 1838627, 1837956, 1956002, 2211492, CNS 2213701, CCF 2217003, DBI 2225775 at Pitt and UMD.

Machine learning tasks with large-scale distributed datasets call for distributed training [3] because of its ability to shorten the calculation time and train models with data from various locations. At the same time, communication overhead has emerged as the most restrictive bottleneck of distributed training due to the increasing model and data size. To tackle these issues, federated learning (FL) [44] has emerged as a promising technique since it makes use of distributed data from different clients, avoids frequent transmission between clients and the central server, and preserves data privacy. In FL, clients train and update their models locally, and the server aggregates and averages the model parameters from all clients periodically. Only models are shared among clients and the training data are stored locally, which provides a certain level of data privacy. In addition, FL also enhances computation power since it utilizes many clients to train models.

Although federated learning has gained popularity, most existing works focus on the standard stochastic minimization problem [33, 48, 34, 36, 71, 58, 70, 72]. Recently, some algorithms for non-minimization optimization in FL are proposed [53, 21, 15, 50, 59]. However, existing FL minimax algorithms have not achieved the complexity level reached by single-machine algorithms. To bridge this gap, we consider the federated nonconvex minimax optimization problem as follows:

$$\min_{x \in \mathbb{R}^{d_1}} \max_{y \in \mathbb{R}^{d_2}} \left\{ F(x,y) = \frac{1}{N} \sum_{i=1}^{N} f_i(x,y) \right\} \tag{1}$$

where the function $f_i(x,y) = \mathbb{E}_{\xi_i \sim \mathcal{D}_i}[f_i(x,y;\xi_i)] : \mathbb{R}^{d_1} \times \mathbb{R}^{d_2} \to \mathbb{R}$ is the loss function of the $i^{th}$ client. We restrict our focus to the non-convex minimax problem, where $f_i(x,y)$ is nonconvex over $x \in \mathbb{R}^{d_1}$ and concave or nonconcave over $y \in \mathbb{R}^{d_2}$. $N$ is the total number of clients. $\xi_i = (x_i, y_i) \sim \mathcal{D}_i$ denotes data point $\xi_i$ is sampled from the local data distribution $\mathcal{D}_i$ on machine $i$. In this paper, heterogeneous datasets are considered, namely, $\mathcal{D}_i$ and $\mathcal{D}_j$ $(i \neq j)$ are not identical.

Some recent works have attempted to solve federated minimax optimization for convex-concave setting [11, 26, 52]. Due to the popularity of deep neural networks, nonconvex minimax has wider applications. More recently, some works [50, 53, 10, 19, 67] extend single-machine algorithms, such as SGDA, to federated learning settings for nonconvex minimax optimization. However, theoretical understandings of federated minimax optimization remain limited in the literature. In the context of stochastic smooth nonconvex minimax problems, their analysis either relies on strict assumptions [19] or achieves suboptimal convergence results [50]. For example, single-machine methods [42, 28] achieve $O(\kappa^3 \varepsilon^{-3})$ under nonconvex-strongly-concave setting, which is much better than $O(\kappa^4 \varepsilon^{-4})$ achieved by the best FL minimax algorithm [50, 63] in existing literature. Therefore, a natural question arises:

> **Can we design stochastic gradient decent ascent methods with better sample and communication complexities to match the convergence rate of single-machine counterparts for solving the problem** (1)**?**

In this paper, we provide an affirmative answer to the aforementioned question and propose a class of algorithms to solve the problem (1) under different settings. In particular, we consider three most common classes of nonconvex minimax optimization problems: 1) NC-C: NonConvex in $x$, Concave in $y$; 2) NC-SC: NonConvex in $x$, Strongly-Concave in $y$; 3) NC- PL: NonConvex in $x$, PL-condition in $y$. For each of these problems, we propose a new algorithm with provably better convergence rate (please see Table 1) and provide a theoretical analysis. Our main contributions are four-fold:

1) NC-C setting. We propose FedSGDA+, and prove it has sample complexity of $O(N^{-1}\varepsilon^{-8})$ and communication complexity of $O(\varepsilon^{-6})$. FedSGDA+ takes advantage of the structure of FL by adding the global learning rate at the server and reduces communication complexity to $O(\varepsilon^{-6})$ from $O(\varepsilon^{-7})$ in [50]. It also achieves a linear speedup to the number of clients.

2) NC-PL setting. We propose a federated stochastic gradient ascent (FedSGDA-M) algorithm with the momentum-based variance reduction technique. It has the best sample complexity of $O(\kappa^3 N^{-1}\varepsilon^{-3})$ and the best communication complexity of $O(\kappa^2 \varepsilon^{-2})$. Compared with existing momentum-based variance reduction algorithms, our result employs a novel theoretical analysis framework that produces a tighter convergence rate (i.e., our rate gets rid of a logarithmic term appearing in existing works).

3) NC-SC setting. FedSGDA-M can be directly applied to the NC-SC setting since the PL condition is weaker than strong-concavity. Our algorithm is the first work to reach sample

Table 1: Complexity comparison of existing nonconvex federated minimax algorithms for finding an $\varepsilon$-stationary point. Sample complexity is the total number of the First-order Oracle (IFO) to reach an $\varepsilon$-stationary point. Communication complexity denotes the total number of back-and-forth communication times between clients and the server. Here, $N$ is the number of clients, and $\kappa = L_f/\mu$ is the condition number.

| Type | Algorithm | Reference | Sample | Communication |
|---|---|---|---|---|
| Nonconvex concave | Local SGDA+ | [50] | $O\left(N^{-1}\varepsilon^{-8}\right)$ | $O(\varepsilon^{-7})$ |
| | FedSGDA+ | Ours | $O\left(N^{-1}\varepsilon^{-8}\right)$ | $O\left(\varepsilon^{-6}\right)$ |
| Nonconvex Strongly Concave | Local SGDA | [50] | $O\left(\kappa^4 N^{-1}\varepsilon^{-4}\right)$ | $O(\kappa^3\varepsilon^{-3})$ |
| | Momentum Local SGDA | [50] | $O\left(\kappa^4 N^{-1}\varepsilon^{-4}\right)$ | $O(\kappa^3\varepsilon^{-3})$ |
| | FEDNEST | [53] | $O\left(\kappa^3\varepsilon^{-4}\right)^a$ | $O\left(\kappa^2\varepsilon^{-4}\right)$ |
| | FedSGDA | Ours | $O\left(\kappa^3 N^{-1}\varepsilon^{-3}\right)$ | $O\left(\kappa^2\varepsilon^{-2}\right)$ |
| Nonconvex PL | Local SGDA | [50] | $O\left(\kappa^4 N^{-1}\varepsilon^{-4}\right)$ | $O(\kappa^3\varepsilon^{-3})$ |
| | Momentum Local SGDA | [50] | $O\left(\kappa^4 N^{-1}\varepsilon^{-4}\right)$ | $O(\kappa^3\varepsilon^{-3})$ |
| | SAGDA | [63] | $O\left(N^{-1}\varepsilon^{-4}\right)$ | $O(\varepsilon^{-2})^b$ |
| | FedSGDA | Ours | $O\left(\kappa^3 N^{-1}\varepsilon^{-3}\right)$ | $O\left(\kappa^2\varepsilon^{-2}\right)$ |

[a] Their theoretical analysis does not report the dependency on N.
[b] Their theoretical analysis does not report the dependency on $\kappa$.

complexity of $O(\varepsilon^{-3})$ in federated learning. In addition, FedSGDA-M does not rely on a large batch size to reach optimal sample complexity compared with single-machine minimax algorithms [29, 42].

4) Extensive experimental results on fair classification and AUROC maximization confirm the effectiveness of our proposed algorithm.

## 2 Related Work

### 2.1 Single-Machine Minimax

**Nonconvex-Concave (NC-C) setting**. [31, 46, 45, 54, 35] proposed various deterministic and stochastic algorithms to solve the NC-C minimax problems. All of these algorithms, however, have a double-loop structure and are thus relatively complicated to implement. They decouple the minimax problem into a minimization problem and a maximization problem and use a nested loop to update variable $y$ while keeping variable $x$ constant. Subsequently, [38] studied the complexity result of the single-loop algorithm (SGDA) for the NC-C minimax problem and proves the stochastic algorithm achieves $O(\varepsilon^{-8})$ complexity. SGDA is a direct extension of SGD from minimization optimization to minimax optimization problems. Recently, [43] providing a unified analysis for the convergence of OGDA and EG methods in the nonconvex-strongly-concave (NC-SC) and nonconvex-concave (NC-C) settings.

**Nonconvex-Strongly-Concave (NC-SC) setting**. [38] analyzed the stochastic gradient descent ascent (SGDA) algorithm and proved that SGDA has $O(\kappa^3\varepsilon^{-4})$ stochastic gradient complexity. To reduce the convergence rate, [42] proposed a stochastic GDA algorithm (i.e. SREDA) with a double-loop structure based on the variance reduction technique of SPIDER [13] and reduce the complexity to $O(\kappa^3\varepsilon^{-3})$. [29] used momentum-based variance reduction technique of STORM [7] and proposed Acc-MDA. Acc-MDA is a single-loop algorithm, which gets the same convergence result as SREDA. Furthermore, adaptive minimax algorithms are introduced [28, 30] to solve the nonsmooth nonconvex-strongly-concave minimax problems based on dynamic mirror functions. [65] used a nested adaptive framework to design parameter-agnostic nonconvex minimax algorithm. [69] proves that VR-based SAPD+ has the complexity of $O(\kappa^2\varepsilon^{-3})$. However, whether the best convergence result of $O(\varepsilon^{-3})$ in single-machine methods can be achieved in the federated setting is an open question. In addition, [23] conducts an in-depth investigation of limitations of GDA algorithm (e.g., smaller learning rate, cycling/divergence issue) and gives a systematic analysis of how to improve GDA dynamics.

**Nonconvex-Nonconcave (NC-NC) setting**. There is extensive research on NC-NC problems [12] and the Nonconvex-PL condition is a special class of functions that interests us the most. Polyak-Łojasiewicz (PL) condition does not require the objective to be concave and recent works show that the PL condition could hold in the training process of overparameterized neural networks with random initialization [1, 5]. Recently, many deterministic methods [45, 64, 14] are proposed for NC-NC problems under the NC-PL setting. [20] proposed a PDAda method for Nonconvex-PL minimax optimization with the restriction of the concavity condition, and [6] focuses on the finite-sum Nonconvex-PL minimax optimization. Stochastic alternating GDA and stochastic smoothed GDA proposed in [66] achieve complexity of $O(\kappa^4 \varepsilon^{-4})$ and $O(\kappa^2 \varepsilon^{-4})$, respectively.

## 2.2 Distributed/Federated Minimax

Distributed training has rapid development in minimax optimization in recent years, driven by the need to train large-scale datasets [40]. Under the serverless decentralized setting, algorithms for nonconvex minimax have been studied extensively in nonconvex-strongly-concave setting [4, 62, 68, 41, 57] and nonconvex-PL setting [27].

In the FL setting, some works analyzed algorithms for convex-concave problems [11, 37, 26, 52]. However, as nonconvex models (e.g., deep neural networks) being more and more prevalent, there is a growing need for federated nonconvex minimax optimization, such as federated adversarial training [49], federated deep AUROC maximization [19] and federated GAN [47]. [19] and [67] focus on imbalanced data tasks. They reformulated the AUROC maximization problem as the minimax problem under the FL setting. But their analysis relies on strict assumptions that deep models satisfy the PL condition and only focuses on PL-strong-concave minimax. [49] converted the robust federated learning into the minimax problem, where only model parameters, namely min variables, are exchanged among clients via the server. [10] proposed Local SGDA and Local SGDA+. Local SGDA is the local-update version of the SGDA algorithm in FL. Different from local SGDA, in local SGDA+, max variable $y$ is updated with a constant min variable $\tilde{x}$ and the snapshot $\tilde{x}$ updates every S iteration. Afterward, [50] improves sample complexity and communication complexity of Local SGDA for NC-SC and NC-PL settings, and Local SGDA+ for NC-C setting. [50] also proposes a Momentum Local SGDA, which achieves the same theoretical results as Local SGDA for NC-PL and NC-SC settings. In addition, [53] designs FEDNEST with two nested loops. Although FEDNEST is composed of FEDINN (a federated stochastic variance reduction algorithm ), their convergence complexity is not improved over vanilla Local SGDA. Afterward, [63] proposes SAGDA under NC-PL setting, which yields a better communication complexity (i.e., $O(\varepsilon^{-2})$). However, its analysis does not consider the effect of condition number $\kappa$. More recently, [51] considers federated minimax optimization with Client Heterogeneity in nonconvex concave and nonconvex strongly-concave settings.

**Relation to Existing Works**. We propose FedSGDA+ for NC-C setting and FedSGDA-M for NC-SC and NC-PL settings. In NC-C setting, we discover that the addition of a global step size leads to better communication complexity. Under this circumstance, theoretical analysis is more challenging as we not only need to consider the complicated structure of the minimax problem but also need to handle the local update and global update separately. For FedSGDA-M, we relax the requirement of step size (specifically designed unnatural step size is often required in STORM-like approaches [7, 34]), which requires novel proof techniques to obtain. Thus our result does not contain a logarithmic term and provides a tighter convergence rate (Seen in contributions 1 in [34]). In addition, with different theoretical frameworks, our better sample complexity doesn't rely on a big batch size, while the single-machine minimax algorithm with variance reduction (Acc-MDA) in [29] (Table 2) needs a large batch to achieve the same sample complexity.

## 3 Algorithms and Convergence Analysis

Notation: $\| \cdot \|$ denotes the $\ell_2$ norm for vectors. $a = O(b)$ denotes that $a \leq Cb$ for some constant $C > 0$. Given the mini-batch samples $\mathcal{B} = \{\xi_j\}_{j=1}^b$, we let $\nabla f_i(x, y; \mathcal{B}) = \frac{1}{b} \sum_{j=1}^b \nabla f_i(x, y; \xi_i)$.

**Assumption 3.1.** (i) Unbiased Gradient. The gradient of each component function $f_i(x, y; \xi)$ computed at each client is unbiased for all $\xi^{(i)} \sim \mathcal{D}_i$, $i \in [N]$:

$$\mathbb{E}[\nabla f_i(x, y; \xi^{(i)})] = \nabla f_i(x, y),$$

**Algorithm 1** FedSGDA+ Algorithm

---

1: **Input:** $T$, local step sizes $\hat{c}, c$, global step sizes $\eta_x, \eta_y$, $k = 0$, numbers of inner updates $Q$ and outer update $S$, and mini-batch size $b$, N clients;
2: **Initialize:** $x_0^i = \tilde{x}_0 = \bar{x}_0, y_0^i = \bar{y}_0$,
3: **for** $t = 0, 1, \ldots, T - 1$ **do**
4:    **for** $i = 1, 2, \ldots, N$ **do**
5:       **Local Update:**
6:       **for** $q = 0, 1, \ldots, Q - 1$ **do**
7:          Draw mini-batch samples $\mathcal{B}_{t,q}^i = \{\xi_i^j\}_{j=1}^b$ with $|\mathcal{B}_t^i| = b$ from $D_i$ locally
8:          $x_{t,q+1}^i = x_{t,q}^i - \hat{c}\nabla_x f_i(x_{t,q}^i, y_{t,q}^i; \mathcal{B}_{t,q}^i)$
9:          $y_{t,q+1}^i = y_{t,q}^i + c\nabla_y f_i(\tilde{x}_k, y_{t,q}^i; \mathcal{B}_{t,q}^i)$
10:       **end for**
11:    **end for**
12:    $x_{t+1,0}^i = \bar{x}_{t+1} = \bar{x}_t + \eta_x \frac{1}{N} \sum_{i=1}^N (x_{t,Q}^i - \bar{x}_t)$
13:    $y_{t+1,0}^i = \bar{y}_{t+1} = \bar{y}_t + \eta_y \frac{1}{N} \sum_{i=1}^N (y_{t,Q}^i - \bar{y}_t)$
14:    **if** $\mod (t + 1, S) = 0$ **then**
15:       $k = k + 1$
16:       $\tilde{x}_k = \bar{x}_{t+1}$
17:    **end if**
18: **end for**
19: **Output:** $x$ and $y$ chosen uniformly random from $\{(\bar{x}_t, \bar{y}_t)\}_{t=1}^T$.

---

(ii) Variance Bound. The following inequalities hold for all $\xi^{(i)} \sim \mathcal{D}_i, i, j \in [N]$:

$$\mathbb{E}\|\nabla f_i(x, y; \xi^{(i)}) - \nabla f_i(x, y)\|^2 \leq \sigma^2$$

$$\frac{1}{N} \sum_{i=1}^N \|\nabla f_i(x, y) - \nabla F(x, y)\|^2 \leq \zeta^2 \tag{2}$$

The Assumption 3.1 is a standard assumption in stochastic optimization, which will be used throughout the rest of the paper. In FL algorithms, the Assumption 3.1 (ii) is frequently used to bound the variance and data heterogeneity. The heterogeneity parameter, $\zeta$, denotes the level of data heterogeneity. In the homogeneous data configuration, $\zeta = 0$.

### 3.1 Nonconvex Concave (NC-C) Problems

**Assumption 3.2.** (Concavity). The nonconvex function $f(\cdot, \cdot)$ is concave in $y$ if for a fixed $x \in \mathbb{R}^{d_1}, \forall y, y' \in \mathbb{R}^{d_2}$, we have

$$f(x, y) \leq f(x, y') + \langle \nabla f(x, y'), y - y' \rangle \tag{3}$$

To solve the problem (1) under the NC-C setting and reduce the complexity, we propose FedSGDA+ (Seen in algorithm 1). Although local SGDA+ [50] achieves the same sample complexity as the single-machine method (SGDA), its communication complexity is not optimal. This unsatisfactory result is due to the fact that local SGDA+ simply extends the single-machine method into the distributed setting, and does not consider the complicated local-server structure in FL.

In FedSGDA+, Line 5-10 are conducted in the local clients. The updates of variable x are similar to the standard stochastic algorithm for minimization problems, such as FedAvg. We sample data points and update the $x$ locally with the current variable $x_{t,q}^i$ and $y_{t,q}^i$. However, for the $y$-updates, stochastic gradients are calculated with the latest snapshot of $x(i.e., \tilde{x}_k)$ and in each local iteration, $y$ updates with the constant $\tilde{x}_k$ as seen in Line 9 in Algorithm 1. The $\tilde{x}_k$ is updated every S rounds ( Line 14-16 in Algorithm 1).

In addition, we make use of the advantage of FL and introduce the global step size $\eta_x, \eta_y$, which provides the flexibility of FL training (Seen in Line 12-13). Local SGDA+ could be regarded as a special case of FedSGDA+. We now provide the convergence analysis of FedSGDA+ and introduce the necessary assumptions. The details of the proofs are provided in the supplementary materials.

**Assumption 3.3.** (Smoothness). Each local function $f_i(x, y)$ has a $L_f$-Lipschitz continuous gradients, i.e., $\forall x_1, x_2$ and $y_1, y_2$, we have

$$\|\nabla f_i(x_1, y_1) - \nabla f_i(x_2, y_2)\| \le L_f \|(x_1, y_1) - (x_2, y_2)\| \tag{4}$$

The Assumption 3.3 on the smoothness is a standard assumption in stochastic optimization [2, 17].

**Assumption 3.4.** (Lipschitz continuity in x). For the function $F$, there exists a constant $G_x$, such that for each $y \in \mathbb{R}^{d_2}$, and $\forall x, x' \in \mathbb{R}^{d_1}$, we have

$$\|F(x, y) - F(x', y)\| \le G_x \|x - x'\|$$

Under the NC-C setting, the function $F(\cdot, \cdot)$ is concave in y. Following [9], we define $\Phi(x) = \max_y F(x, y)$ and the Moreau envelope of $\Phi(\cdot)$ is defined below:

**Definition 3.5.** (Moreau Envelope) A function $\Phi_\lambda(\cdot)$ is the $\lambda$-Moreau envelope of $\Phi(\cdot)$, for $\lambda > 0$, if $\forall x \in R^{d_1}$,

$$\Phi_\lambda(x) = \min_z \Phi(z) + \frac{1}{2\lambda}\|z - x\|^2$$

From [38], we know that when we have a point $x$ that is an $\varepsilon$-stationary point of $\Phi_\lambda(x)$, then $x$ is close to a point $x'$ which is stationary for $\Phi(x)$. Hence, we focus on minimizing $\|\nabla\Phi_\lambda(x)\|$ under the NC-C setting.

**Theorem 3.6.** *Suppose Assumptions 3.1, 3.2, 3.3, 3.4 hold and the sequences $\{x_t, y_t\}$ are generated by Algorithm 1, $\max\{c\eta_y, c\} \le \frac{1}{10QL_f}$ and let $\|\bar{y}_t\|^2 \le D$ following [10, 50],*

$$\frac{1}{T}\sum_{t=0}^{T-1} \mathbb{E}\left\|\nabla\Phi_{1/2L_f}(\bar{x}_t)\right\|^2 \le 8L_f\hat{c}\eta_x(QG_x^2 + \frac{\sigma^2}{N}) + 8\frac{\mathbb{E}\left[\Phi_{1/2L_f}(x_0)\right] - \mathbb{E}\left[\Phi_{1/2L_f}(\bar{x}_T)\right]}{Q\hat{c}\eta_x T}$$

$$+48L_f^2Q[\hat{c}^2 + c^2](\sigma^2 + 6Q\zeta^2) + 288L_f^2Q^2\hat{c}^2G_x^2$$

$$+576L_f^3Q^2c^2[\frac{D}{c\eta_yQS} + \frac{c\eta_y\sigma^2}{N} + 6L_fQ^2c^2(\sigma^2 + 6\zeta^2)]$$

$$+32L_fG_x\eta_x\hat{c}SQ\sqrt{G_x^2 + \frac{\sigma^2}{N}} + \frac{16L_fD}{c\eta_yQS} + \frac{16L_f(c\eta_y)\sigma^2}{N} + 96L_f^2Q^2c^2(\sigma^2 + 6\zeta^2)]$$

**Corollary 3.7.** *By setting $c = \hat{c} = \frac{1}{10L_fQT^{1/3}}$, $Q = \frac{T^{1/3}}{N}$, $\hat{c}\eta_x = \frac{N}{10L_fT}$, $c\eta_y = \frac{1}{10L_fQ} = \frac{N}{10L_fT^{1/3}}$, $S = T^{1/3}$, FedSGDA+ has the following convergence rate:*

$$\frac{1}{T}\sum_{t=0}^{T-1} \mathbb{E}\left\|\nabla\Phi_{1/2L_f}(\bar{x}_t)\right\|^2 \le \frac{80L_f\Delta}{T^{1/3}} + \frac{4(G_x^2 + \sigma^2)}{5T^{2/3}} + \frac{24(\sigma^2 + 6\zeta^2)}{25T^{2/3}} + \frac{72G_x^2}{25T^{2/3}} + \frac{24(\sigma^2 + 6\zeta^2)}{25T^{2/3}}$$

$$+ \frac{144L_f}{25T^{2/3}}[\frac{10L_fD}{T^{1/3}} + \frac{\sigma^2}{10L_fT^{1/3}} + \frac{3(\sigma^2 + 6\zeta^2)}{50L_fT^{2/3}}] + \frac{16G_x}{5T^{1/3}}\sqrt{G_x^2 + \frac{\sigma^2}{N}} + \frac{160L_f^2D}{T^{1/3}} + \frac{8\sigma^2}{5T^{1/3}}$$

*Remark* 3.8. (Complexity) Based on Corollary 3.7, to make $\frac{1}{T}\sum_{t=0}^{T-1}\mathbb{E}\left\|\nabla\Phi_{1/2L_f}(\bar{x}_t)\right\|^2 \le \varepsilon^2$, communication complexity $T = O(\varepsilon^{-6})$. We choose $b = O(1)$, then we have sample complexity $bQT = N^{-1}\varepsilon^{-8}$. It also denotes that FedSGDA+ has linear speedup with respect to the number of clients.

## 3.2 Nonconvex-PL (NC-PL) Problems

**Assumption 3.9.** (Polyak Łojasiewicz (PL) Condition in y). The function $F(x, y)$ is $\mu$-PL condition in $y(\mu > 0)$, if $\forall x$: 1) $y^*(x) = \arg\max_y F(x, y)$ has a nonempty solution set; 2) $\|\nabla_y F(x, y)\|^2 \ge 2\mu(F(x, y^*(x)) - F(x, y)), \forall y$.

To solve problem (1) with better convergence complexity under nonconvex-PL, we propose federated stochastic gradient ascent (FedSGDA-M) algorithm with the momentum-based variance reduction technique (Seen in in algorithm 2).

**Algorithm 2** FedSGDA-M Algorithm

---

1: **Input:** $T$, step sizes $\hat{c}, c, \eta$; momentum coefficient $\alpha, \beta$, the number of local updates $Q$, and mini-batch size $b$ and initial mini-batch size $B$;
2: **Initialize:** $x_0^i = \bar{x}_0 = \frac{1}{N}\sum_{i=1}^N x_0^i, y_0^i = \bar{y}_0 = \frac{1}{N}\sum_{i=1}^N y_0^i$, $u_1^i = \nabla_x f(x_0^i, y_0^i; \mathcal{B}_0^i)$ and $v_1^i = \nabla_y f(x_0^i, y_0^i; \mathcal{B}_0^i)$ where $|\mathcal{B}_0^i| = B$ are drawn from $D_i$ for $i \in [N]$.
3: **for** $t = 1, 2, \ldots, T$ **do**
4:     **for** $i = 1, 2, \ldots, N$ **do**
5:         **if** $\mod (t, Q) = 0$ **then**
6:             **Sever Update:**
7:             $u_t^i = \bar{u}_t = \frac{1}{N}\sum_{j=1}^N u_t^j$
8:             $v_t^i = \bar{v}_t = \frac{1}{N}\sum_{j=1}^N v_t^j$
9:             $x_t^i = \bar{x}_t = \frac{1}{N}\sum_{j=1}^N (x_{t-1}^j - \hat{c}\eta u_t^j)$
10:            $y_t^i = \bar{y}_t = \frac{1}{N}\sum_{j=1}^N (y_{t-1}^j + c\eta v_t^j)$
11:         **else**
12:             $x_t^i = x_{t-1}^i - \hat{c}\eta u_t^i$
13:             $y_t^i = y_{t-1}^i + c\eta v_t^i$
14:         **end if**
15:         Draw mini-batch samples $\mathcal{B}_t^i = \{\xi_i^j\}_{j=1}^b$ with $|\mathcal{B}_t^i| = b$ from $D_i$ locally
16:         $u_{t+1}^i = \nabla_x f_i(x_t^i, y_t^i; \mathcal{B}_t^i) + (1-\alpha)(u_t^i - \nabla_x f_i(x_{t-1}^i, y_{t-1}^i; \mathcal{B}_t^i))$
17:         $v_{t+1}^i = \nabla_y f_i(x_t^i, y_t^i; \mathcal{B}_t^i) + (1-\beta)(v_t^i - \nabla_y f_i(x_{t-1}^i, y_{t-1}^i; \mathcal{B}_t^i))$
18:     **end for**
19: **end for**
20: **Output:** $x$ and $y$ chosen uniformly random from $\{\bar{x}_t, \bar{y}_t\}_{t=1}^T$.

---

In FedSGDA-M, each client initializes the gradient estimators $\{u_1^i, v_1^i\}$ with stochastic gradient as seen in line 2 of Algorithm 2. Following that, each client updates the model variables $\{x_t^i, y_t^i\}$ locally as standard stochastic gradient descent ascent method (lines 12-13 of Algorithm 2). Compared with local momentum SGDA [50], the key difference is that clients utilize variance reduction gradient estimators $\{u_t^i, v_t^i\}$, which are constructed in lines 15-17 of Algorithm 2. For the update step of $\{u_t, v_t\}$, the coefficients should satisfy $0 < \alpha < 1$ and $0 < \beta < 1$. In every $Q$ iteration, clients transmit model parameters and gradient estimators to the server, which computes $\{\bar{x}_t, \bar{y}_t, \bar{u}_t, \bar{v}_t\}$. Then the server sends averaged model variables and gradient estimators to each client to update the local variables, as shown in lines 5-10 of Algorithm 2.

**Definition 3.10.** According to Assumption 3.9, there exists at least one solution to the problem $\max_y F(x, y)$ for any $x$. Here we define $\Phi(x) = F(x, y^*(x)) = \max_y F(x, y)$. We use $\varepsilon$-stationary point of $\Phi(x)$, i.e. $\|\nabla\Phi(x)\| \leq \varepsilon$ as the convergence metric.

We know $\Phi(x)$ is differentiable and $(L + \kappa L)$-smooth and $y^*(\cdot)$ is $\kappa$-Lipschitz from [50]. Given that $\nabla_y F(\bar{x}_t, y^*(x_t)) = 0$, we have $\nabla\Phi(\bar{x}_t) = \nabla_x F(\bar{x}_t, y^*(x_t)) + \nabla_y F(\bar{x}_t, y^*(x_t)) \cdot \partial y^*(\bar{x}_t) = \nabla_x F(\bar{x}_t, y^*(x_t))$ which is widely used in the analysis of nonconvex-PL [10] and nonconvex-strongly-concave minimax optimization [62, 38]. Then we discuss the convergence analysis of FedSGDA-M. The proofs are provided in the supplementary materials.

**Assumption 3.11.** (Lipschitz Smoothness) Each component function $f_i(x, y; \xi)$ has a $L_f$-Lipschitz gradient, i.e., $\forall x_1, x_2$ and $y_1, y_2$, we have

$$\mathbb{E}\|\nabla f_i(x_1, y_1; \xi) - \nabla f_i(x_2, y_2; \xi)\| \leq L_f\|(x_1, y_1) - (x_2, y_2)\| \tag{5}$$

$F(x, y)$ has an $L_f$-Lipschitz gradient based on the convexity of norm and Assumption 3.11. We have

$$\|\nabla_x F(x_1, y_1) - \nabla_x F(x_2, y_2)\| = \left\|\frac{1}{N}\sum_{i=1}^N \mathbb{E}[\nabla_x f_i(x_1, y_1; \xi) - \nabla_x f_i(x_2, y_2; \xi)]\right\|$$

$$\leq \frac{1}{N}\sum_{i=1}^N \mathbb{E}\|\nabla_x f_i(x_1, y_1; \xi) - \nabla_x f_i(x_2, y_2; \xi)\|$$

$$\leq L_f\|(x_1, y_1) - (x_2, y_2)\|$$

In optimization analysis, it is standard to use the Assumption 3.11. Several widely used single-machine stochastic algorithms, such as SPIDER [13] and STORM [7], make use of this assumption. It is also used by numerous FL algorithms, including MIME [32] Fed-GLOMO [8], STEM [34] and FAFED [58].

**Theorem 3.12.** *Suppose that sequence $\{\bar{x}_t, \bar{y}_t\}_{t=0}^T$ is generated from Algorithm 2. Under the above Assumptions (3.1,3.9,3.11), given $\eta = \frac{1}{20QL}$, $\alpha = c_1\eta^2$, $\beta = c_2\eta^2$, $c_1 = \frac{30L^2}{bN\kappa^{1-\nu}}$, $c_2 = \frac{30L^2}{bN\kappa^{2-2\nu}}$, $c = \frac{1}{6\kappa^{1-\nu}}$, $\hat{c} = \frac{1}{54\kappa^{3-\nu}}$ where $\nu \in [0,1]$ we have*

$$\frac{1}{T}\sum_{t=0}^{T-1}\mathbb{E}\|\nabla\Phi(\bar{x}_t)\|^2 \leq \frac{2[\Phi(\bar{x}_0) - \Phi(\bar{x}_T)]}{\hat{c}\eta T} + \frac{3\sigma^2}{\alpha TBN} + \frac{36L_f^2\sigma^2}{\mu^2\beta TBN} + \frac{12L_f^2}{c\eta\mu^2 T}[\Phi(\bar{x}_0) - F(\bar{x}_0, \bar{y}_t)]$$

$$+ \frac{6\alpha\sigma^2}{Nb} + \frac{72\beta\sigma^2 L_f^2}{Nb\mu^2} + \left[\frac{\sigma^2(c_1^2 + c_2^2)}{30bL^2} + \frac{\zeta^2(c_1^2 + c_2^2)}{12L^2}\right]\kappa^2\eta^2$$

**Corollary 3.13.** *By setting $b = O(\kappa^\nu)$ for $\nu \in [0,1]$, $c_1 = \frac{30L^2}{bN\kappa^{1-\nu}}$, $c_2 = \frac{30L^2}{bN\kappa^{2-2\nu}}$, $c = \frac{1}{6\kappa^{1-\nu}}$, $\hat{c} = \frac{1}{54\kappa^{3-\nu}}$, $T = \kappa^{3-\nu}T_0$, $Q = \frac{T_0^{1/3}}{N^{2/3}}$, $\eta = \frac{1}{20QL} = \frac{N^{2/3}}{20LT_0^{1/3}}$, $B = \frac{T_0^{1/3}b\kappa^{1-\nu}}{N^{2/3}}$, we have $\alpha = c_1\eta^2 = \frac{3N^{1/3}}{40T_0^{2/3}b\kappa^{1-\nu}}$, $\beta = c_2\eta^2 = \frac{3N^{1/3}}{40T_0^{2/3}b\kappa^{2-2\nu}}$.*

$$\frac{1}{T}\sum_{t=0}^{T-1}\mathbb{E}\|\nabla\Phi(\bar{x}_t)\|^2 \leq \frac{2160L[\Phi(\bar{x}_0) - \Phi^*]}{(NT_0)^{2/3}} + \frac{40\sigma^2}{\kappa^{3-\nu}(NT_0)^{2/3}} + \frac{480\sigma^2}{\kappa^2(NT_0)^{2/3}}$$

$$+ \frac{240L_f}{(NT_0)^{2/3}}[\Phi(\bar{x}_0) - F(\bar{x}_0, \bar{y}_0)] + \frac{9\sigma^2}{20b\kappa(NT_0)^{2/3}} + \frac{27\sigma^2}{5(NT_0)^{2/3}} + \left[\frac{3\sigma^2}{20b} + \frac{15\zeta^2}{40}\right]\frac{1}{(NT_0)^{2/3}}$$

*where $\Phi^*$ is the optimal.*

*Remark* 3.14. (Complexity) To make the $\frac{1}{T}\sum_{t=0}^{T-1}\mathbb{E}\|\nabla\Phi(\bar{x}_t)\|^2 \leq \varepsilon^2$, we get $T_0 = O(N^{-1}\varepsilon^{-3})$ and $T = O(\kappa^{3-\nu}N^{-1}\varepsilon^{-3})$. Considering the $b = \kappa^\nu$, we have communication complexity $\frac{T}{Q} = \kappa^{3-\nu}(NT_0)^{2/3} = \kappa^{3-\nu}\varepsilon^{-2}$ and sample complexity $bT = O(\kappa^3 N^{-1}\varepsilon^{-3})$. When $\nu = 1, b = \kappa$, communication Complexity $\frac{T}{Q} = \kappa^2\varepsilon^{-2}$ for finding an $\varepsilon$-stationary point. Sample complexity $bT = O(\kappa^3 N^{-1}\varepsilon^{-3})$ matches the complexity result achieved by the single-machine algorithms, such as SREDA and Acc-MDA in [42, 29] but we do not require a large batch size b compared with these algorithms. And $O(\kappa^3 N^{-1}\varepsilon^{-3})$ also exhibits a linear speed-up compared with the aforementioned single-machine algorithms.

## 3.3 Nonconvex-Strongly-Concave (NC-SC) Problems

**Assumption 3.15.** Each local function function $f_i(x, y)$ is $\mu$-strongly concave in $y \in \mathcal{Y}$, i.e., $\forall x \in \mathcal{X}$ and $y_1, y_2 \in \mathcal{Y}$, we have

$$\|\nabla_y f_i(x, y_1) - \nabla_y f_i(x, y_2)\| \geq \mu\|y_1 - y_2\|$$

When the function $F(x, y)$ is strongly concave in $y \in \mathcal{Y}$, there exists a unique solution to the problem $\max_{y \in \mathcal{Y}} f(x, y)$ for any $x$. Since PL condition is weaker than strong concavity, the convergence result of FedSGDA-M in Algorithm 2 under NC-PL also apply to NC-SC problem and FedSGDA-M has the sample complexity of $O(\kappa^3 n^{-1}\varepsilon^{-3})$ and the communication complexity of $O(\kappa^2\varepsilon^{-2})$ under nonconvex-strongly-concave setting.

## 4 Experiments

We conduct experiments on AUROC maximization and fair classification tasks to verify the efficiency of our algorithms under nonconvex-strongly-concave and nonconvex-concave settings. Experiments are completed on the computer cluster with AMD EPYC 7513 Processors and NVIDIA RTX A6000. The code is available [*]

---

[*]https://github.com/xidongwu/Federated-Minimax-and-Conditional-Stochastic-Optimization

**Datasets and Models**: We test the performance of algorithms on three typical datasets: Fashion-MNIST dataset, CIFAR-10 dataset and Tiny-ImageNet. Fashion-MNIST dataset has $70,000$ $28 \times 28$ gray images (10 categories, $60,000$ training images and $10,000$ testing images). CIFAR-10 dataset consists of $50,000$ training images and $10,000$ testing images. Each image is the $3 \times 32 \times 32$ arrays of color image. Tiny-ImageNet dataset has 200 classes of $(64 \times 64)$ colored images and each class has 500 training images, 50 validation images, and 50 test images. For Fashion MNIST and Cifar10 data sets, we choose convolutional neural network from [28] (The details are shown in the supplementray materials). For Tiny-ImageNet, we choose ResNet-18 [24] as the model.

## 4.1 Fair Classification

First, we follow [50, 28] and train fair classification networks by minimizing the maximum loss over different categories.

$$\min_{x} \max_{y \in \mathcal{Y}} \frac{1}{N} \sum_{i=1}^{N} \sum_{c=1}^{C} y_c \mathcal{L}_c^i(x) \quad \text{s.t.} \mathcal{Y} = \left\{ y \mid y_i \geq 0, \sum_{i=1}^{C} y_i = 1 \right\}$$

where $\mathcal{L}_c^i$ denotes the cross-entropy loss functions corresponding to the class $c$ in C different classes and $x$ denotes the CNN model parameters. Clearly, the problem in (6) is nonConvex in x (deep model parameters), and Concave in y. We compare our algorithm (FedSGDA+) and local SGDA+ with varying model, datasets, local update numbers, step size. Although the constraint is not considered in the theoretical analysis of local SGDA+ and FedSGDA+, FedSGDA+ still shows better performance compared with local SGDA+.

The network has 20 clients. The datasets are partitioned into disjoint sets across all clients and each client holds part of the data from all the classes [34]. We initialize the renset18 with pre-trained weights in PyTorch. In experiments, we run grid search for step size, and choose the step size for primal variable in the set $\{0.01, 0.03, 0.05, 0.1, 0.3\}$ and that for dual variable in the set $\{0.001, 0.01, 0.1\}$. We choose the global step size in the set $\{0.1, 0.5, 1, 1.5, 2\}$. The batch-size $b$ is in 50 and the inner loop number $Q$ is seleted from $\{20, 50, 100\}$, The outer loop number $S$ is selected from $\{1, 5, 10\}$ for FedSGDA+ and $\{1, 5, 10, Q\}$ for local SGDA+.

Figure 1 shows that FedSGDA+ has a better convergence rate than local SGDA+. This confirms that our algorithm can effectively accelerate SGDA by using the structure of federated learning. Due to the page limitition, the ablation analysis of step size is presented in the supplementary materials.

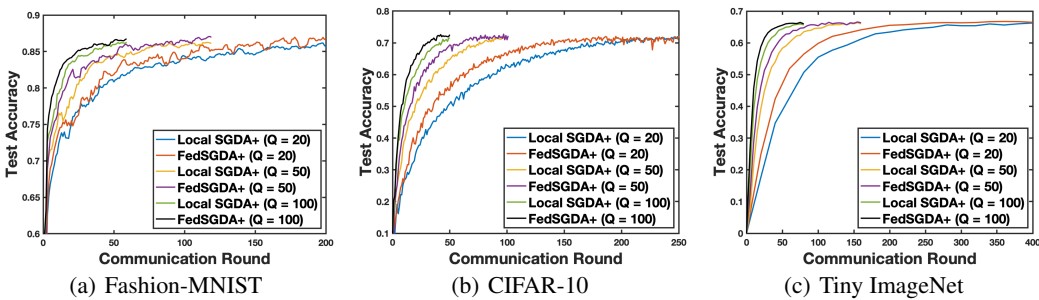

| (a) Fashion-MNIST | (b) CIFAR-10 | (c) Tiny ImageNet |

Figure 1: Test Accuracy vs the number of communication rounds during the training phase.

## 4.2 AUROC Maximization

[39] showed the AUROC maximization problem could be reformulated as the non-convex-strongly-concave minimax optimization, as below:

$$\min_{\substack{\mathbf{m} \in \mathbb{R}^d \\ (a,b) \in \mathbb{R}^2}} \max_{w \in \mathbb{R}} \frac{1}{N} \sum_{i=1}^{N} \mathbb{E}_{\xi_i \sim \mathcal{D}_i} \left[ f_i \left( \mathbf{m}, a, b, w; \xi \right) \right] \tag{6}$$

where

$$f\left(\mathbf{m}, a, b, w; \xi\right) = (1-p)\left(h\left(\mathbf{m}; \mathbf{x}\right) - a\right)^2 \mathbb{I}_{[y=1]} + p\left(h\left(\mathbf{m}; \mathbf{x}\right) - b\right)^2 \mathbb{I}_{[y=-1]}$$
$$+ 2(1+w)[ph\left(\mathbf{m}; \mathbf{x}\right)\mathbb{I}_{[y=-1]} - (1-p)h\left(\mathbf{m}, \mathbf{x}\right)\mathbb{I}_{[y=1]}] - p(1-p)w^2$$

$\xi_i = (\mathbf{x}, y) \sim \mathcal{D}_i$ denotes a random data point and $\mathbf{x}$ represents the data features and $y \in \mathcal{Y} = \{-1, +1\}$ is the label. $h(\mathbf{m}; \mathbf{x})$ denotes the prediction score of the data point $\mathbf{x}$ calculated by a model with parameter $\mathbf{m}$. $p = Pr(y=1) = \mathbb{E}_y[\mathbb{I}_{[y=1]}]$ denotes the prior probability of the positive data.

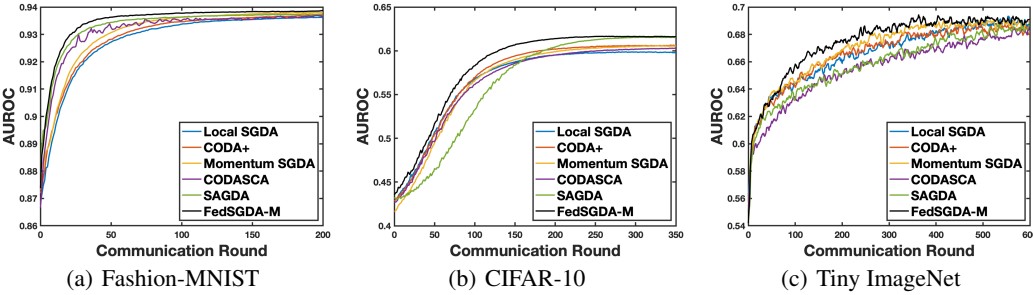

(a) Fashion-MNIST        (b) CIFAR-10        (c) Tiny ImageNet

Figure 2: AUROC scores on the test datasets vs the number of communication rounds during the training phase.

Following [19, 67], we constructed the imbalanced binary-class versions of datasets as follow: firstly, the first half of the classes (0 - 4) in the original Fashion-MNIST, CIFAR10 and classes (0 - 99) in Tiny-ImageNet datasets are converted into the negative class, and the rest half of classes are considered to be a positive class. 80% of the negative data points are randomly dropped in each dataset. Then the datasets are evenly divided into disjoint sets across 16 clients. In this case, each clients share completely different imbalanced datasets. In the experiment, we use xavier normal initialization to deep models.

We compare our algorithm (i.e., FedSGDA-M) with local SGDA [10, 50], CODA+ [19, 67], Momentum SGDA [50], CODASCA [67] and SAGDA [63] as baselines in AUROC maximization. In experiments, we carefully tune hyperparameters for all methods. We run a grid search for step size, and choose the step size for the primal variable in the set $\{0.001, 0.005, 0.01\}$ and that for dual variable in the set $\{0.0001, 0.001, 0.01\}$. We choose the global step size from $\{0.9, 1, 1.5, 2\}$ for CODASCA and SAGDA. We choose the momentum parameter in Local Momentum SGDA in the set $\{0.1, 0.5, 0.9\}$. The $\alpha$ and $\beta$ in FedSGDA-M are chosen from $\{0.1, 0.5, 0.9\}$. The batch-size $b$ is 50 and the inner loop number $Q \in \{10, 20, 50\}$.

As shown in Figure 2, we compare the performance of FedSGDA-M and other baseline methods against the number of communication rounds. Figure 2 shows that our algorithms consistently outperform the other baseline algorithms on testing datasets, which validates the efficacy of our algorithms. Due to space limitation, other test results are provided in the supplementary materials.

**Limitation**. Minimax optimization has many applications and a more comprehensive discussion of our proposed algorithms on these tasks will be a future study because the theoretical analysis is the main contribution of this paper.

## 5 Conclusion

In this paper, we study a class of federated nonconvex minimax optimization problems (1). We consider the three most common settings (NC-SC, NC-PL, NC-C). Under the NC-C setting, we propose FedSGDA+ and prove it has the best communication complexity of $O(\varepsilon^{-6})$. It also achieves a linear speedup to the number of clients. Under NC-PL and NC-PL settings, we propose FedSGDA-M with variance reduction technique and we prove that our algorithm (FedSGDA-M) has the best sample complexity ($O(\kappa^3 n^{-1}\varepsilon^{-3})$) and the best sample communication complexity ($O(\kappa^2 \varepsilon^{-2})$). We prove that FedSGDA also enjoys linear speedup with respect to the number of clients. Therefore, we reduce the existing complexity results for the most common nonconvex minimax optimization problems under the federated learning setting.

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
