# A Preliminary Results

**Lemma A.1.** *[45, 10]*

*1) Under the above Assumptions 3.9 and 3.11, the function $\Phi(x) = \max_y f(x,y) = f(x, y^*(x))$ and the mapping $y^*(x) = \arg\max_y f(x,y)$ have L-Lipschitz continuous gradient and $\kappa$-Lipschitz continuous respectively, such as for all $x_1, x_2 \in \mathbb{R}^{d_1}$*

$$\|\nabla\Phi(x_1) - \nabla\Phi(x_2)\| \leq L\|x_1 - x_2\|, \quad \|y^*(x_1) - y^*(x_2)\| \leq \kappa\|x_1 - x_2\|, \tag{7}$$

*where $L = L_f(1 + \kappa)$ and $\kappa = L_f/\mu$.*

*2) (Quadratic growth condition) If the function $g(x)$ satisfies Assumptions 3.9 and 3.11, then $\forall x$, the following conditions holds*

$$g(x) - \min_z g(z) \geq \frac{\mu}{2} \|x_p - x\|^2 \tag{8}$$

$$\|\nabla g(x)\|^2 \geq 2\mu \left( g(x) - \min_z g(z) \right) \tag{9}$$

*3) If F is L–smooth then*

$$\|\nabla_y F(x,y)\|^2 \leq 2L_f[\Phi(x) - F(x,y)] \tag{10}$$

**Lemma A.2.** *[34] For a finite sequence $x_t^n \in \mathbb{R}^d$, $n \in [N]$ and $\bar{x}_t \in \mathbb{R}^d$, we have*

$$\sum_{n=1}^{N} \|x_t^n - \bar{x}_t\|^2 \leq \sum_{n=1}^{N} \|x_t^n\|^2 \tag{11}$$

# B Nonconvex-Concave (NS-C)

In this part, we analyze the convergence result of FedSGDA+ in Algorithm 1. We define $\hat{x}_t = \arg\min_x \Phi(x) + L_f \|x - \bar{x}_t\|^2$, and define $\Phi_{1/2L_f}(\bar{x}_t) = \min_x \Phi(x) + L_f \|x - \bar{x}_t\|^2$

## B.1 Important Conclusions

**Lemma B.1.** *Suppose Assumptions 3.1, 3.2, 3.3, 3.4 hold and the sequences $\{x_t, y_t\}$ are generated by Algorithm 1, we have*

$$\mathbb{E}\left[\Phi_{1/2L_f}(\bar{x}_{t+1})\right] \leq \mathbb{E}\left[\Phi_{1/2L_f}(\bar{x}_t)\right] + QL_f(\hat{c}\eta_x)^2(QG_x^2 + \frac{\sigma^2}{N}) - \frac{Q\hat{c}\eta_x}{8}\mathbb{E}\left\|\nabla\Phi_{1/2L_f}(\bar{x}_t)\right\|^2$$

$$+ \frac{2\hat{c}\eta_x L_f^2}{N} \sum_{i=1}^{N} \sum_{q=0}^{Q-1} \mathbb{E}(\|x_{t,q}^i - \bar{x}_t\|^2 + \|y_{t,q}^i - \bar{y}_t\|^2) + 2Q\hat{c}\eta_x L_f \mathbb{E}\left[\Phi(\bar{x}_t) - F(\bar{x}_t, \bar{y}_t)\right]$$

*Proof.* Based on the $x$ update step in Algorithm 1, we have

$$\mathbb{E}\left\|\hat{x}_t - \bar{x}_{t+1}\right\|^2 = \mathbb{E}\left\|\hat{x}_t - \bar{x}_t + \frac{\hat{c}\eta_x}{N}\sum_{i=1}^{N}\sum_{q=0}^{Q-1}\nabla_x f_i\left(x_{t,q}^i, y_{t,q}^i; \mathcal{B}_{t,q}^i\right)\right\|^2$$

$$= \mathbb{E}\left\|\hat{x}_t - \bar{x}_t\right\|^2 + (\hat{c}\eta_x)^2\mathbb{E}\left\|\frac{1}{N}\sum_{i=1}^{N}\sum_{q=0}^{Q-1}\nabla_x f_i\left(x_{t,q}^i, y_{t,q}^i; \mathcal{B}_{t,q}^i\right)\right\|^2$$

$$+ 2\hat{c}\eta_x\mathbb{E}\left\langle \hat{x}_t - \bar{x}_t, \frac{1}{N}\sum_{i=1}^{N}\sum_{q=0}^{Q-1}\left[\nabla_x f_i\left(x_{t,q}^i, y_{t,q}^i\right) - \nabla_x F\left(\bar{x}_t, \bar{y}_t\right)\right]\right\rangle$$

$$+ 2\hat{c}\eta_x\mathbb{E}\left\langle \hat{x}_t - \bar{x}_t, Q\nabla_x F\left(\bar{x}_t, \bar{y}_t\right)\right\rangle$$

$$\overset{(a)}{\leq}\mathbb{E}\left\|\hat{x}_t - \bar{x}_t\right\|^2 + (\hat{c}\eta_x)^2\mathbb{E}\left[\left\|\frac{1}{N}\sum_{i=1}^{N}\sum_{q=0}^{Q-1}\nabla_x f_i\left(x_{t,q}^i, y_{t,q}^i\right)\right\|^2\right.$$

$$+ \left.\left\|\frac{1}{N}\sum_{i=1}^{N}\sum_{q=0}^{Q-1}[\nabla_x f_i\left(x_{t,q}^i, y_{t,q}^i\right) - \nabla_x f_i\left(x_{t,q}^i, y_{t,q}^i; \mathcal{B}_{t,q}^i\right)]\right\|^2\right]$$

$$+ \hat{c}\eta_x\mathbb{E}\left[\frac{QL_f}{2}\left\|\hat{x}_t - \bar{x}_t\right\|^2 + \frac{2}{QL_f}\left\|\frac{1}{N}\sum_{i=1}^{N}\sum_{q=0}^{Q-1}[\nabla_x f_i\left(x_{t,q}^i, y_{t,q}^i\right) - \nabla_x F\left(\bar{x}_t, \bar{y}_t\right)]\right\|^2\right]$$

$$+ 2\hat{c}\eta_x Q\mathbb{E}\left\langle \hat{x}_t - \bar{x}_t, \nabla_x F\left(\bar{x}_t, \bar{y}_t\right)\right\rangle$$

$$\overset{(b)}{\leq}\mathbb{E}\left\|\hat{x}_t - \bar{x}_t\right\|^2 + Q(\hat{c}\eta_x)^2\left(QG_x^2 + \frac{\sigma^2}{N}\right) + \frac{Q\hat{c}\eta_x L_f}{2}\left\|\hat{x}_t - \bar{x}_t\right\|^2$$

$$+ \frac{2\hat{c}\eta_x L_f}{N}\sum_{i=1}^{N}\sum_{q=0}^{Q-1}\mathbb{E}(\|x_{t,q}^i - \bar{x}_t\|^2 + \|y_{t,q}^i - \bar{y}_t\|^2) + 2Q\hat{c}\eta_x\mathbb{E}\left\langle \hat{x}_t - \bar{x}_t, \nabla_x F\left(\bar{x}_t, \bar{y}_t\right)\right\rangle \quad (12)$$

where (a) follows from Assumption 3.1 (i) unbiasedness of stochastic gradients and Young's inequality; (b) follows from Assumption 3.4 to bound the gradient and 3.1 (ii) bound the variance and the smooth of $f_i(x, y)$ in Assumption 3.3 . Next, we bound the last term in (12).

$$\mathbb{E}\left\langle \hat{x}_t - \bar{x}_t, \nabla_x F\left(\bar{x}_t, \bar{y}_t\right)\right\rangle \overset{(a)}{\leq}\mathbb{E}\left[F\left(\hat{x}_t, \bar{y}_t\right) - F\left(\bar{x}_t, \bar{y}_t\right) + \frac{L_f}{2}\left\|\hat{x}_t - \bar{x}_t\right\|^2\right]$$

$$\overset{(b)}{\leq}\mathbb{E}\left[\Phi\left(\hat{x}_t\right) - F\left(\bar{x}_t, \bar{y}_t\right) + \frac{L_f}{2}\left\|\hat{x}_t - \bar{x}_t\right\|^2\right]$$

$$= \mathbb{E}\left[\Phi\left(\hat{x}_t\right) + L_f\left\|\hat{x}_t - \bar{x}_t\right\|^2\right] - \mathbb{E}F\left(\bar{x}_t, \bar{y}_t\right) - \frac{L_f}{2}\mathbb{E}\left\|\hat{x}_t - \bar{x}_t\right\|^2$$

$$\overset{(c)}{\leq}\mathbb{E}\left[\Phi\left(\bar{x}_t\right) + L_f\left\|\bar{x}_t - \bar{x}_t\right\|^2\right] - \mathbb{E}F\left(\bar{x}_t, \bar{y}_t\right) - \frac{L_f}{2}\mathbb{E}\left\|\hat{x}_t - \bar{x}_t\right\|^2$$

$$= \mathbb{E}\left[\Phi\left(\bar{x}_t\right) - F\left(\bar{x}_t, \bar{y}_t\right) - \frac{L_f}{2}\left\|\hat{x}_t - \bar{x}_t\right\|^2\right] \quad (13)$$

where (a) holds by $L_f$-smoothness of $f(x, \cdot)$ (Assumption 3.3); (b) uses the definition of $\Phi(x)$; (c) uses the definition of $\hat{x}$. We also have

$$\Phi_{1/2L_f}\left(\bar{x}_{t+1}\right) = \min_x \Phi(x) + L_f\left\|x - \bar{x}_{t+1}\right\|^2 \leq \Phi\left(\hat{x}_t\right) + L_f\left\|\hat{x}_t - \bar{x}_{t+1}\right\|^2 \quad (14)$$

Combining the above (12), (13) and (14), we get

$$\mathbb{E}\left[\Phi_{1/2L_f}\left(\bar{x}_{t+1}\right)\right]$$

$$\leq \mathbb{E}\Phi\left(\hat{x}_t\right) + L_f\left[\mathbb{E}\left\|\hat{x}_t - \bar{x}_t\right\|^2 + Q(\hat{c}\eta_x)^2(QG_x^2 + \frac{\sigma^2}{N})\right] + \frac{Q\hat{c}\eta_x L_f^2}{2}\left\|\hat{x}_t - \bar{x}_t\right\|^2$$

$$+ \frac{2\hat{c}\eta_x L_f^2}{N}\sum_{i=1}^{N}\sum_{q=0}^{Q-1}\mathbb{E}(\|x_{t,q}^i - \bar{x}_t\|^2 + \|y_{t,q}^i - \bar{y}_t\|^2) + 2Q\hat{c}\eta_x L_f\mathbb{E}\left[\Phi\left(\bar{x}_t\right) - F\left(\bar{x}_t, \bar{y}_t\right) - \frac{L_f}{2}\|\hat{x}_t - \bar{x}_t\|^2\right]$$

$$\leq \mathbb{E}\left[\Phi_{1/2L_f}\left(\bar{x}_t\right)\right] + QL_f(\hat{c}\eta_x)^2(QG_x^2 + \frac{\sigma^2}{N}) + \frac{2\hat{c}\eta_x L_f^2}{N}\sum_{i=1}^{N}\sum_{q=0}^{Q-1}\mathbb{E}(\|x_{t,q}^i - \bar{x}_t\|^2 + \|y_{t,q}^i - \bar{y}_t\|^2)$$

$$- \frac{Q\hat{c}\eta_x L_f^2}{2}\mathbb{E}\|\hat{x}_t - \bar{x}_t\|^2 + 2Q\hat{c}\eta_x L_f\mathbb{E}\left[\Phi\left(\bar{x}_t\right) - F\left(\bar{x}_t, \bar{y}_t\right)\right]$$

$$\overset{(a)}{=} \mathbb{E}\left[\Phi_{1/2L_f}\left(\bar{x}_t\right)\right] + QL_f(\hat{c}\eta_x)^2(QG_x^2 + \frac{\sigma^2}{N}) + \frac{2\hat{c}\eta_x L_f^2}{N}\sum_{i=1}^{N}\sum_{q=0}^{Q-1}\mathbb{E}(\|x_{t,q}^i - \bar{x}_t\|^2 + \|y_{t,q}^i - \bar{y}_t\|^2)$$

$$- \frac{Q\hat{c}\eta_x}{8}\mathbb{E}\left\|\nabla\Phi_{1/2L_f}\left(\bar{x}_t\right)\right\|^2 + 2Q\hat{c}\eta_x L_f\mathbb{E}\left[\Phi\left(\bar{x}_t\right) - F\left(\bar{x}_t, \bar{y}_t\right)\right]$$

where (a) holds due to the fact $\nabla\Phi_{1/2L_f}(\hat{x}_t) = 2L_f(\hat{x}_t - \bar{x}_t)$ from Lemma 2.2 in [9]     □

**Lemma B.2.** *Suppose Assumptions 3.1, 3.2, 3.3, 3.4 hold and the sequences $\{x_t, y_t\}$ are generated by Algorithm 1, $c \leq \frac{1}{10QL_f}$ and $\hat{c} \leq \frac{1}{10QL_f}$, we have*

$$\sum_{i=1}^{N}\sum_{q=0}^{Q-1}\mathbb{E}\left[\left\|x_{t,q}^i - \bar{x}_t\right\|^2 + \left\|y_{t,q}^i - \bar{y}_t\right\|^2\right]$$

$$\leq \sum_{i=1}^{N}\left[3Q^2(\hat{c}^2 + c^2)(\sigma^2 + 6Q\zeta^2) + 18Q^3\hat{c}^2G_x^2 + 36L_fQ^3c^2\mathbb{E}[\Phi(\tilde{x}_k) - F(\tilde{x}_k, \bar{y}_t)]\right]$$

*Proof.*

$$\mathbb{E}\left\|x_{t,q}^i - \bar{x}_t\right\|^2 = \mathbb{E}\left\|x_{t,Q-1}^i - \bar{x}_t - \hat{c}\nabla_x f_i(x_{t,Q-1}^i, y_{t,Q-1}^i; \mathcal{B}_{t,Q-1}^i)\right\|^2$$

$$\leq \mathbb{E}\|x_{t,Q-1}^i - \bar{x}_t - \hat{c}[\nabla_x f_i(x_{t,Q-1}^i, y_{t,Q-1}^i; \mathcal{B}_{t,Q-1}^i) - \nabla_x f_i(x_{t,Q-1}^i, y_{t,Q-1}^i) + \nabla_x f_i(x_{t,Q-1}^i, y_{t,Q-1}^i)$$

$$- \nabla_x f_i(\bar{x}_t, \bar{y}_t) + \nabla_x f_i(\bar{x}_t, \bar{y}_t) - \nabla_x f(\bar{x}_t, \bar{y}_t) + \nabla_x f(\bar{x}_t, \bar{y}_t)]\|^2$$

$$\leq (1 + \frac{1}{2Q-1})\mathbb{E}\left\|x_{t,Q-1}^i - \bar{x}_t\right\|^2 + \mathbb{E}\left\|\hat{c}(\nabla_x f_i(x_{t,Q-1}^i, y_{t,Q-1}^i; \mathcal{B}_{t,Q-1}^i) - \nabla_x f_i(x_{t,Q-1}^i, y_{t,Q-1}^i))\right\|^2$$

$$+ 6Q\mathbb{E}\left\|\hat{c}\left(\nabla_x f_i(x_{t,Q-1}^i, y_{t,Q-1}^i) - \nabla_x f_i(\bar{x}_t, \bar{y}_t)\right)\right\|^2 + 6Q\mathbb{E}\|\hat{c}\left(\nabla_x f_i(\bar{x}_t, \bar{y}_t) - \nabla_x F(\bar{x}_t, \bar{y}_t)\right)\|^2$$

$$+ 6Q\|\hat{c}\nabla F\left(\bar{x}_t, \bar{y}_t\right)\|^2$$

$$\sum_{i=1}^{N}\mathbb{E}\left\|x_{t,q}^i - \bar{x}_t\right\|^2 \leq \sum_{i=1}^{N}\left[(1 + \frac{1}{2Q-1})\mathbb{E}\left\|x_{t,Q-1}^i - \bar{x}_t\right\|^2 + \hat{c}^2\sigma^2\right.$$

$$+ 6Q\hat{c}^2L_f^2\mathbb{E}\left[\left\|x_{t,Q-1}^i - \bar{x}_t\right\|^2 + \left\|y_{t,Q-1}^i - \bar{y}_t\right\|^2\right] + 6Q\hat{c}^2\zeta^2 + 6Q\hat{c}^2\|\nabla_x F\left(\bar{x}_t, \bar{y}_t\right)\|^2\right] \quad (15)$$

The error bound $\left\|y_{t,q}^i - \bar{y}_t\right\|^2$ follow a similar analysis but y updates with a fixed $\tilde{x}_k$,

$$\sum_{i=1}^{N}\mathbb{E}\left\|y_{t,q}^i - \bar{y}_t\right\|^2 = \sum_{i=1}^{N}\mathbb{E}\left[\left\|y_{t,Q-1}^i - \bar{y}_t - c\nabla_y f_i(\tilde{x}_k, y_{t,Q-1}^i; \mathcal{B}_{t,Q-1}^i)\right\|^2\right]$$

$$\leq \sum_{i=1}^{N}\left[(1 + \frac{1}{2Q-1})\mathbb{E}\left\|y_{t,Q-1}^i - \bar{y}_t\right\|^2 + c^2\sigma^2 + 6Qc^2L_f^2\mathbb{E}\left\|y_{t,Q-1}^i - \bar{y}_t\right\|^2 + 6Qc^2\zeta^2\right]$$

$$+ 6Qc^2\mathbb{E}\|\nabla_y F\left(\tilde{x}_k, \bar{y}_t\right)\|^2\right] \quad (16)$$

and we can get

$$\sum_{i=1}^{N} \mathbb{E}\left[\left\|x_{t,q}^{i} - \bar{x}_t\right\|^2 + \left\|y_{t,q}^{i} - \bar{y}_t\right\|^2\right]$$

$$\leq \sum_{i=1}^{N}\left[\left(1 + \frac{1}{2Q-1} + 6Q\hat{c}^2 L_f^2 + 6Qc^2 L_f^2\right) \mathbb{E}\left[\left\|x_{t,Q-1}^{i} - \bar{x}_t\right\|^2 + \left\|\bar{y}_{t,Q-1}^{i} - \bar{y}_t\right\|^2\right] + [\hat{c}^2 + c^2]\sigma^2\right.$$

$$\left. + 6Q[\hat{c}^2 + c^2]\zeta^2 + 6Q\hat{c}^2 \mathbb{E}\|\nabla_x F\left(\bar{x}_t, \bar{y}_t\right)\|^2 + 6Qc^2 \mathbb{E}\|\nabla_y F\left(\tilde{x}_k, \bar{y}_t\right)\|^2\right]$$

$$\leq \sum_{i=1}^{N}\left[\left(1 + \frac{1}{Q-1}\right) \mathbb{E}\left[\left\|\bar{x}_{t,Q-1}^{i} - \bar{x}_t\right\|^2 + \left\|\bar{y}_{t,Q-1}^{i} - \bar{y}_t\right\|^2\right] + (\hat{c}^2 + c^2)\sigma^2 + 6Q(\hat{c}^2 + c^2)\zeta^2\right.$$

$$\left. + 6Q\hat{c}^2 \mathbb{E}\|\nabla_x F\left(\bar{x}_t, \bar{y}_t\right)\|^2 + 6Qc^2 \mathbb{E}\|\nabla_y F\left(\tilde{x}_k, \bar{y}_t\right)\|^2\right]$$

where the last inequality follows from the fact that $\frac{1}{4Q-2} + 6Q(\hat{c})^2 L^2 \leq \frac{1}{2Q-2}$ if $(\hat{c})^2 \leq \frac{1}{6(2(Q)^2-3Q+1)L^2}$. Similarly, $\frac{1}{4Q-2} + 6Q(c)^2 L^2 \leq \frac{1}{2Q-2}$ if $(c)^2 \leq \frac{1}{6(2(Q)^2-3Q+1)L^2}$. Unrolling the recursion, we obtain:

$$\sum_{i=1}^{N} \mathbb{E}\left[\left\|x_{t,q}^{i} - \bar{x}_t\right\|^2 + \left\|y_{t,q}^{i} - \bar{y}_t\right\|^2\right]$$

$$\leq \sum_{i=1}^{N}\sum_{q=0}^{Q-1}\left(1 + \frac{1}{Q-1}\right)^q \left[(\hat{c}^2 + c^2)\sigma^2 + 6Q(\hat{c}^2 + c^2)\zeta^2 + 6Q\hat{c}^2 \mathbb{E}\|\nabla_x F\left(\bar{x}_t, \bar{y}_t\right)\|^2\right.$$

$$\left. + 6Qc^2 \mathbb{E}\|\nabla_y F\left(\bar{x}_t, \bar{y}_t\right)\|^2\right]\right]$$

$$\leq \sum_{i=1}^{N}(Q-1)\left[\left(1 + \frac{1}{Q-1}\right)^Q - 1\right]\left[(\hat{c}^2 + c^2)(\sigma^2 + 6Q\zeta^2) + 6Q\hat{c}^2 \mathbb{E}\|\nabla_x F\left(\bar{x}_t, \bar{y}_t\right)\|^2\right.$$

$$\left. + 6Qc^2 \mathbb{E}\|\nabla_y F\left(\bar{x}_t, \bar{y}_t\right)\|^2\right]\right]$$

$$\leq \sum_{i=1}^{N}\left[3Q(\hat{c}^2 + c^2)(\sigma^2 + 6Q\zeta^2) + 18Q^2\hat{c}^2 \mathbb{E}\|\nabla_x F\left(\bar{x}_t, \bar{y}_t\right)\|^2 + 18Q^2 c^2 \mathbb{E}\|\nabla_y F\left(\tilde{x}_k, \bar{y}_t\right)\|^2\right]$$

$$\tag{17}$$

where the third inequality holds due to $(1 + \frac{1}{Q-1})^{Q-1} \leq e \leq 3$. Furthermore, based on assumption Assumption 3.4 and Lemma A.1 (3), we have

$$\mathbb{E}\|\nabla_x F\left(\bar{x}_t, \bar{y}_t\right)\|^2 \leq G_x^2$$

$$\mathbb{E}\|\nabla_y F\left(\tilde{x}_k, \bar{y}_t\right)\|^2 \leq 2L_f \mathbb{E}[\Phi(\tilde{x}_k) - F(\tilde{x}_k, \bar{y}_t)]$$

Putting , we get the final result.

$$\sum_{i=1}^{N}\sum_{q=0}^{Q-1} \mathbb{E}\left[\left\|x_{t,q}^{i} - \bar{x}_t\right\|^2 + \left\|y_{t,q}^{i} - \bar{y}_t\right\|^2\right]$$

$$\leq \sum_{i=1}^{N}\left[3Q^2(\hat{c}^2 + c^2)(\sigma^2 + 6Q\zeta^2) + 18Q^3\hat{c}^2 G_x^2 + 36L_f Q^3 c^2 \mathbb{E}[\Phi(\tilde{x}_k) - F(\tilde{x}_k, \bar{y}_t)]\right]$$

$\square$

**Lemma B.3.** *Suppose Assumptions 3.1, 3.2, 3.3, 3.4 hold and the sequences $\{\bar{x}_t, \bar{y}_t\}$ are generated by Algorithm 1, $\max\{c\eta_y, c\} \le \frac{1}{10QL_f}$ and let $\|\bar{y}_t\|^2 \le D$, we have*

$$\frac{1}{S} \sum_{t=kS}^{(k+1)S-1} \mathbb{E}\left[\Phi\left(\bar{x}_t\right) - F\left(\bar{x}_t, \bar{y}_t\right)\right]$$

$$\le 2G_x\eta_x\hat{c}(S-1)Q\sqrt{G_x^2 + \frac{\sigma^2}{N}} + \frac{D}{c\eta_y QS} + \frac{c\eta_y\sigma^2}{N} + [2(c\eta_y)QL_f^2 + L_f][3Q(c)^2(\sigma^2 + 6Q\zeta^2)]$$

*Proof.* When $t = kS$ to $(k+1)S - 1$, where $k$ is a positive integer. Let $\tilde{\mathbf{x}}_k$ is the latest snapshot iterate in Algorithm 1. Then

$$\begin{aligned}
&\mathbb{E}\left[\Phi\left(\bar{x}_t\right) - F\left(\bar{x}_t, \bar{y}_t\right)\right] \\
=&\mathbb{E}\left[F\left(\bar{x}_t, y^*\left(\bar{x}_t\right)\right) - F\left(\tilde{x}_k, y^*\left(\tilde{x}_k\right)\right)\right] + \mathbb{E}\left[F\left(\tilde{x}_k, y^*\left(\tilde{x}_k\right)\right) - F\left(\tilde{x}_k, \bar{y}_t\right)\right] + \mathbb{E}\left[F\left(\tilde{x}_k, \bar{y}_t\right) - F\left(\bar{x}_t, \bar{y}_t\right)\right] \\
\le&2G_x\mathbb{E}\left\|\tilde{x}_k - \bar{x}_t\right\| + \mathbb{E}\left[F\left(\tilde{x}_k, y^*\left(\tilde{x}_k\right)\right) - F\left(\tilde{x}_k, \bar{y}_t\right)\right] \\
=&2G_x\mathbb{E}\left\|\tilde{x}_k - \bar{x}_t\right\| + \mathbb{E}\left[\Phi\left(\tilde{x}_k\right) - F\left(\tilde{x}_k, \bar{y}_t\right)\right]
\end{aligned} \tag{18}$$

where the last inequality follows from $G_x$-Lipschitz continuity of $F(\cdot, y)$ (Assumption 3.4). For the first term, we have

$$\begin{aligned}
2G_x\mathbb{E}\left\|\bar{x}_t - \tilde{x}_k\right\| =&2G_x\mathbb{E}\|\frac{\hat{c}\eta_x}{N} \sum_{t=kS+1}^{(k+1)S-1} \sum_{q=0}^{Q-1} \sum_{i=1}^{N} \nabla_x f_i\left(x_{t,q}^i, y_{t,q}^i; \mathcal{B}_{t,q}^i\right)\| \\
=&2G_x\hat{c}\eta_x \sum_{t=kS+1}^{(k+1)S-1} \sum_{q=0}^{Q-1} \mathbb{E}\|\frac{1}{N} \sum_{i=1}^{N} \nabla_x f_i\left(x_{t,q}^i, y_{t,q}^i; \mathcal{B}_{t,q}^i\right)\| \\
\le&2G_x\eta_x\hat{c}(S-1)Q\sqrt{G_x^2 + \frac{\sigma^2}{N}}
\end{aligned} \tag{19}$$

Next, we bound the second term in (18). During the updates of $\{y_t^i\}$, from $t = kS$ to $(k+1)S - 1$, the variable $\tilde{x}_k$ keep constant. The update of variable $y$ likes maximizing a concave function $f(\tilde{x}_k, \cdot)$ in the federated learning setting with local update as Q. Based on update step of variable $y$, we have

$$\begin{aligned}
\|\bar{y}_{t+1} - y^*(\tilde{x}_k)\|^2 =&\left\| \bar{y}_t + c\eta_y \frac{1}{N} \sum_{i=1}^{N} \sum_{q=0}^{Q-1} \nabla_y f_i\left(\tilde{x}_k, y_{t,q}^i; \mathcal{B}_{t,q}^i\right) - y^*(\tilde{x}_k) \right\|^2 \\
=&\|\bar{y}_t - y^*(\tilde{x}_k)\|^2 + (c\eta_y)^2 \left\| \frac{1}{N} \sum_{i=1}^{N} \sum_{q=0}^{Q-1} \nabla_y f_i\left(\tilde{x}_k, y_{t,q}^i; \mathcal{B}_{t,q}^i\right) \right\|^2 \\
&+ 2c\eta_y \left\langle \bar{y}_t - y^*(\tilde{x}_k), \frac{1}{N} \sum_{i=1}^{N} \sum_{q=0}^{Q-1} \nabla_y f_i\left(\tilde{x}_k, y_{t,q}^i; \mathcal{B}_{t,q}^i\right) \right\rangle
\end{aligned}$$

Taking expectation on the both side and we have

$$\begin{aligned}
\mathbb{E}\|\bar{y}_{t+1} - y^*(\tilde{x}_k)\|^2 =&\mathbb{E}\|\bar{y}_t - y^*(\tilde{x}_k)\|^2 + (c\eta_y)^2\mathbb{E}\left\| \frac{1}{N} \sum_{i=1}^{N} \sum_{q=0}^{Q-1} \nabla_y f_i\left(\tilde{x}_k, y_{t,q}^i; \mathcal{B}_{t,q}^i\right) \right\|^2 \\
&+ 2c\eta_y\mathbb{E} \left\langle \bar{y}_t - y^*(\tilde{x}_k), \frac{1}{N} \sum_{i=1}^{N} \sum_{q=0}^{Q-1} \nabla_y f_i\left(\tilde{x}_k, y_{t,q}^i\right) \right\rangle
\end{aligned} \tag{20}$$

Let $r_t = \bar{y}_t - y^*(\tilde{x}_k)$, and using Lemma A.1 (3) that

$$\mathbb{E}\|\nabla_y F\left(\tilde{x}_k, \bar{y}_t\right)\| \le 2L_f\mathbb{E}[\Phi(\tilde{x}_k) - F(\tilde{x}_k, \bar{y}_t)] \tag{21}$$

$$\mathbb{E}\left\|r_{t+1}\right\|^2$$

$$\leq \|r_t\|^2 + (c\eta_y)^2 \mathbb{E}\|\frac{1}{N}\sum_{i=1}^{N}\sum_{q=0}^{Q-1}[\nabla_y f_i\left(\tilde{x}_k, y_{t,q}^i; \mathcal{B}_{t,q}^i\right) - \nabla_y f_i\left(\tilde{x}_k, y_{t,q}^i\right) + \nabla_y f_i\left(\tilde{x}_k, y_{t,q}^i\right)$$

$$- \nabla_y f_i\left(\tilde{x}_k, \bar{y}_t\right) + \nabla_y f_i\left(\tilde{x}_k, \bar{y}_t\right)\|^2 + 2c\eta_y \mathbb{E}\left\langle \bar{y}_t - y^*(\tilde{x}_k), \frac{1}{N}\sum_{i=1}^{N}\sum_{q=0}^{Q-1}\nabla_y f_i\left(\tilde{x}_k, y_{t,q}^i\right)\right\rangle$$

$$\leq \|r_t\|^2 + (c\eta_y)^2 \mathbb{E}\|\frac{1}{N}\sum_{i=1}^{N}\sum_{q=0}^{Q-1}[\nabla_y f_i\left(\tilde{x}_k, y_{t,q}^i; \mathcal{B}_{t,q}^i\right) - \nabla_y f_i\left(\tilde{x}_k, y_{t,q}^i\right)\|^2$$

$$+ 2(c\eta_y)^2 \mathbb{E}\|\frac{1}{N}\sum_{i=1}^{N}\sum_{q=0}^{Q-1}[\nabla_y f_i\left(\tilde{x}_k, y_{t,q}^i\right) - \nabla_y f_i\left(\tilde{x}_k, \bar{y}_t\right)]\|^2 + 2(c\eta_y)^2 \mathbb{E}\|\frac{1}{N}\sum_{i=1}^{N}\sum_{q=0}^{Q-1}\nabla_y f_i\left(\tilde{x}_k, \bar{y}_t\right)\|^2$$

$$+ 2c\eta_y \frac{1}{N}\sum_{i=1}^{N}\sum_{q=0}^{Q-1}\mathbb{E}\left\langle \bar{y}_t - y_{t,q}^i, \nabla_y f_i\left(\tilde{x}_k, y_{t,q}^i\right)\right\rangle + 2c\eta_y \frac{1}{N}\sum_{i=1}^{N}\sum_{q=0}^{Q-1}\mathbb{E}\left\langle y_{t,q}^i - y^*(\tilde{x}_k), \nabla_y f_i\left(\tilde{x}_k, y_{t,q}^i\right)\right\rangle$$

$$\overset{(a)}{\leq} \|r_t\|^2 + \frac{(c\eta_y)^2\sigma^2 Q}{N} + \frac{2(c\eta_y)^2 Q L_f^2}{N}\sum_{i=1}^{N}\sum_{q=0}^{Q-1}\mathbb{E}\|y_{t,q}^i - \bar{y}_t\|^2 + 4(c\eta_y)^2 Q^2 L_f \mathbb{E}[\Phi(\tilde{x}_k) - F(\tilde{x}_k, \bar{y}_t)]$$

$$+ 2c\eta_y \frac{1}{N}\sum_{i=1}^{N}\sum_{q=0}^{Q-1}\mathbb{E}[f_i\left(\tilde{x}_k, \bar{y}_t\right) - f_i\left(\tilde{x}_k, y_{t,q}^i\right) + \frac{L_f}{2}\|\bar{y}_t - y_{t,q}^i\|^2 + f_i\left(\tilde{x}_k, y_{t,q}^i\right) - f_i\left(\tilde{x}_k, y^*(\tilde{x}_k)\right)]$$

$$= \|r_t\|^2 + \frac{(c\eta_y)^2\sigma^2 Q}{N} + \frac{2(c\eta_y)^2 Q L_f^2}{N}\sum_{i=1}^{N}\sum_{q=0}^{Q-1}\mathbb{E}\|y_{t,q}^i - \bar{y}_t\|^2 + 4(c\eta_y)^2 Q^2 L_f \mathbb{E}[\Phi(\tilde{x}_k) - F(\tilde{x}_k, \bar{y}_t)]$$

$$+ 2c\eta_y \frac{1}{N}\sum_{i=1}^{N}\sum_{q=0}^{Q-1}\mathbb{E}[F\left(\tilde{x}_k, \bar{y}_t\right) + \frac{L_f}{2}\|\bar{y}_t - y_{t,q}^i\|^2 - F\left(\tilde{x}_k, y^*(\tilde{x}_k)\right)]$$

$$= \|r_t\|^2 + \frac{(c\eta_y)^2\sigma^2 Q}{N} + \frac{2(c\eta_y)^2 Q L_f^2}{N}\sum_{i=1}^{N}\sum_{q=0}^{Q-1}\mathbb{E}\|y_{t,q}^i - \bar{y}_t\|^2 + 4(c\eta_y)^2 Q^2 L_f \mathbb{E}[\Phi(\tilde{x}_k) - F(\tilde{x}_k, \bar{y}_t)]$$

$$+ c\eta_y L_f \frac{1}{N}\sum_{i=1}^{N}\sum_{q=0}^{Q-1}\mathbb{E}\|\bar{y}_t - y_{t,q}^i\|^2 - 2c\eta_y Q\mathbb{E}[\Phi\left(\tilde{x}_k\right) - F\left(\tilde{x}_k, \bar{y}_t\right)]$$

$$\leq \|r_t\|^2 + \frac{(c\eta_y)^2\sigma^2 Q}{N} + \frac{2(c\eta_y)^2 Q L_f^2 + c\eta_y L_f}{N}\sum_{i=1}^{N}\sum_{q=0}^{Q-1}\mathbb{E}\|y_{t,q}^i - \bar{y}_t\|^2$$

$$- 2c\eta_y Q[1 - 2c\eta_y Q L_f]\mathbb{E}[\Phi(\tilde{x}_k) - F(\tilde{x}_k, \bar{y}_t)]$$

$$\overset{(b)}{\leq} \|r_t\|^2 + \frac{(c\eta_y)^2\sigma^2 Q}{N} + [2(c\eta_y)^2 Q L_f^2 + c\eta_y L_f][3Q^2(c)^2(\sigma^2 + 6Q\zeta^2)]$$

$$- 2c\eta_y Q[1 - 2c\eta_y Q L_f - 36 L_f^3 Q^3(c\eta_y)c^2 - 18 L_f^2 Q^2 c^2]\mathbb{E}[\Phi(\tilde{x}_k) - F(\tilde{x}_k, \bar{y}_t)]$$

$$\overset{(c)}{\leq} \|r_t\|^2 + \frac{(c\eta_y)^2\sigma^2 Q}{N} + [2(c\eta_y)^2 Q L_f^2 + c\eta_y L_f][3Q^2 c^2(\sigma^2 + 6Q\zeta^2)]$$

$$- c\eta_y Q\mathbb{E}[\Phi(\tilde{x}_k) - F(\tilde{x}_k, \bar{y}_t)] \tag{22}$$

where (a) holds due to Lemma A.1 (3) that

$$\mathbb{E}\|\nabla_y F\left(\tilde{x}_k, \bar{y}_t\right)\| \leq 2L_f \mathbb{E}[\Phi(\tilde{x}_k) - F(\tilde{x}_k, \bar{y}_t)]$$

and the smoothness of $f_i(\tilde{x}_k, \cdot)$

$$\mathbb{E}\left\langle \bar{y}_t - y_{t,q}^i, \nabla_y f_i\left(\tilde{x}_k, y_{t,q}^i\right)\right\rangle \leq f_i\left(\tilde{x}_k, \bar{y}_t\right) - f_i\left(\tilde{x}_k, y_{t,q}^i\right) + \frac{L_f}{2}\|\bar{y}_t - y_{t,q}^i\|^2$$

and concave property that

$$\mathbb{E}\left\langle y_{t,q}^i - y^*(\tilde{x}_k), \nabla_y f_i\left(\tilde{x}_k, y_{t,q}^i\right)\right\rangle \le f_i\left(\tilde{x}_k, y_{t,q}^i\right) - f_i\left(\tilde{x}_k, y^*(\tilde{x}_k)\right)$$

and (b) holds due to the (16) and following the analysis in (17), we have

$$\frac{1}{N}\sum_{i=1}^{N}\sum_{q=0}^{Q-1}\mathbb{E}\left\|y_{t,q}^i - \bar{y}_t\right\|^2 \le 3Q^2c^2(\sigma^2 + 6Q\zeta^2) + 36L_f Q^3 c^2 \mathbb{E}[\Phi(\tilde{x}_k) - F(\tilde{x}_k, \bar{y}_t)]$$

and (c) holds due to $\max\{c\eta_y, c\} \le \frac{1}{10QL_f}$ and $[1 - 2c\eta_y QL_f - 36L_f^3 Q^3(c\eta_y)(c)^2 - 18L_f^2 Q^2(c)^2] \ge \frac{1}{2}$

Furthermore, rearranging the terms in (22), and summing from $t = kS$ to $(k+1)S - 1$, we have

$$\frac{1}{S}\sum_{t=kS}^{(k+1)S-1}\mathbb{E}\left[\Phi\left(\tilde{x}_k\right) - f\left(\tilde{x}_k, \bar{y}_t\right)\right]$$

$$\le \frac{1}{S}\sum_{t=kS}^{(k+1)S-1}\left[\frac{\|r_t\|^2 - \|r_{t+1}\|^2}{c\eta_y Q} + \frac{(c\eta_y)\sigma^2}{N} + [2(c\eta_y)QL_f^2 + L_f][3Qc^2(\sigma^2 + 6Q\zeta^2)]\right] \quad (23)$$

Putting (19) and (23) into (18), let $\|\bar{y}_t\|^2 \le D$ for all t. we have

$$\frac{1}{S}\sum_{t=kS}^{(k+1)S-1}\mathbb{E}\left[\Phi\left(\bar{x}_t\right) - F\left(\bar{x}_t, \bar{y}_t\right)\right]$$

$$\le 2G_x\eta_x\hat{c}(S-1)Q\sqrt{G_x^2 + \frac{\sigma^2}{N}} + \frac{1}{S}\sum_{t=kS}^{(k+1)S-1}\mathbb{E}\left[\Phi\left(\tilde{x}_k\right) - F\left(\tilde{x}_k, \bar{y}_t\right)\right]$$

$$\le 2G_x\eta_x\hat{c}(S-1)Q\sqrt{G_x^2 + \frac{\sigma^2}{N}} + \frac{D}{c\eta_y QS} + \frac{c\eta_y\sigma^2}{N} + [2(c\eta_y)QL_f^2 + L_f][3Q(c)^2(\sigma^2 + 6Q\zeta^2)]$$

$\square$

## B.2 Proof of Theorem

In this part, we show the Proof of Theorem 3.6.

*Proof.* Recall Lemma B.1

$$\mathbb{E}\left[\Phi_{1/2L_f}\left(\bar{x}_{t+1}\right)\right] \le \mathbb{E}\left[\Phi_{1/2L_f}\left(\bar{x}_t\right)\right] + QL_f(\hat{c}\eta_x)^2\left(QG_x^2 + \frac{\sigma^2}{N}\right) - \frac{Q\hat{c}\eta_x}{8}\mathbb{E}\left\|\nabla\Phi_{1/2L_f}\left(\bar{x}_t\right)\right\|^2$$

$$+ \frac{2\hat{c}\eta_x L_f^2}{N}\sum_{i=1}^{N}\sum_{q=0}^{Q-1}\mathbb{E}(\|x_{t,q}^i - \bar{x}_t\|^2 + \|y_{t,q}^i - \bar{y}_t\|^2) + 2Q\hat{c}\eta_x L_f\mathbb{E}\left[\Phi\left(\bar{x}_t\right) - F\left(\bar{x}_t, \bar{y}_t\right)\right]$$

$$\frac{1}{T}\sum_{t=0}^{T-1}\mathbb{E}\left\|\nabla\Phi_{1/2L_f}\left(\bar{x}_t\right)\right\|^2 \leq 8\frac{\mathbb{E}\left[\Phi_{1/2L_f}\left(\bar{x}_0\right)\right]-\mathbb{E}\left[\Phi_{1/2L_f}\left(\bar{x}_T\right)\right]}{Q\hat{c}\eta_x T}+8L_f(\hat{c}\eta_x)(QG_x^2+\frac{\sigma^2}{N})$$

$$+\frac{16L_f^2}{QNT}\sum_{t=0}^{T-1}\sum_{i=1}^{N}\sum_{q=0}^{Q-1}\mathbb{E}(\|x_{t,q}^i-\bar{x}_t\|^2+\|y_{t,q}^i-\bar{y}_t\|^2)+\frac{16L_f}{T}\sum_{t=0}^{T-1}\mathbb{E}\left[\Phi\left(\bar{x}_t\right)-F\left(\bar{x}_t,\bar{y}_t\right)\right]$$

$$\overset{(a)}{\leq} 8\frac{\mathbb{E}\left[\Phi_{1/2L_f}\left(\bar{x}_0\right)\right]-\mathbb{E}\left[\Phi_{1/2L_f}\left(\bar{x}_T\right)\right]}{Q\hat{c}\eta_x T}+8L_f(\hat{c}\eta_x)(QG_x^2+\frac{\sigma^2}{N})$$

$$+48L_f^2 Q[\hat{c}^2+c^2](\sigma^2+6Q\zeta^2)+288L_f^2 Q^2\hat{c}^2 G_x^2$$

$$+\frac{576L_f^3 Q^2 c^2}{T}\sum_{k=0}^{T/S}\sum_{t=kS}^{(k+1)S-1}\mathbb{E}[\Phi(\tilde{x}_k)-F(\tilde{x}_k,\bar{y}_t)]+\frac{16L_f}{T}\sum_{t=0}^{T-1}\mathbb{E}\left[\Phi\left(\bar{x}_t\right)-F\left(\bar{x}_t,\bar{y}_t\right)\right]$$

$$\overset{(b)}{\leq} 8\frac{\mathbb{E}\left[\Phi_{1/2L_f}\left(\bar{x}_0\right)\right]-\mathbb{E}\left[\Phi_{1/2L_f}\left(\bar{x}_T\right)\right]}{Q\hat{c}\eta_x T}+8L_f(\hat{c}\eta_x)(QG_x^2+\frac{\sigma^2}{N})$$

$$+48L_f^2 Q[\hat{c}^2+c^2](\sigma^2+6Q\zeta^2)+288L_f^2 Q^2\hat{c}^2 G_x^2$$

$$+576L_f^3 Q^2 c^2[\frac{D}{c\eta_y QS}+\frac{(c\eta_y)\sigma^2}{N}+6L_f Q^2 c^2(\sigma^2+6\zeta^2)]$$

$$+32L_f G_x\eta_x\hat{c}SQ\sqrt{G_x^2+\frac{\sigma^2}{N}}+\frac{16L_f D}{c\eta_y QS}+\frac{16L_f(c\eta_y)\sigma^2}{N}+96L_f^2 Q^2 c^2(\sigma^2+6\zeta^2)]$$

where (a) holds due to Lemma B.2; (b) is using the (23) and Lemma B.3.

Let $c=\hat{c}=\frac{1}{10L_f QT^{1/3}}$, $Q=\frac{T^{1/3}}{N}$, $\hat{c}\eta_x=\frac{N}{10L_f T}$, $c\eta_y=\frac{1}{10L_f Q}=\frac{N}{10L_f T^{1/3}}$, $S=T^{1/3}$

$$\frac{1}{T}\sum_{t=0}^{T-1}\mathbb{E}\left\|\nabla\Phi_{1/2L_f}\left(\bar{x}_t\right)\right\|^2$$

$$\leq\frac{80L_f\Delta}{T^{1/3}}+\frac{4(G_x^2+\sigma^2)}{5T^{2/3}}+\frac{24(\sigma^2+6\zeta^2)}{25T^{2/3}}+\frac{72G_x^2}{25T^{2/3}}$$

$$+\frac{144L_f}{25T^{2/3}}[\frac{10L_f D}{T^{1/3}}+\frac{\sigma^2}{10L_f T^{1/3}}+\frac{3(\sigma^2+6\zeta^2)}{50L_f T^{2/3}}]$$

$$+\frac{16G_x}{5T^{1/3}}\sqrt{G_x^2+\frac{\sigma^2}{N}}+\frac{160L_f^2 D}{T^{1/3}}+\frac{8\sigma^2}{5T^{1/3}}+\frac{24(\sigma^2+6\zeta^2)}{25T^{2/3}} \quad (24)$$

To make $\frac{1}{T}\sum_{t=0}^{T-1}\mathbb{E}\left\|\nabla\Phi_{1/2L_f}\left(\bar{x}_t\right)\right\|^2\leq\varepsilon^2$, we have communication complexity $T=O(\varepsilon^{-6})$. Set $b=O(1)$, $QT=N^{-1}\varepsilon^{-8}$ $\qquad\square$

## C  FedSGDA-M in Nonconvex-PL (NC-PL) setting

In this section, we provide the detailed convergence analysis of FedSGDA-M. For convenience, in the subsequent analysis, we define $f_{t,i}=f_i(x_t^i,y_t^i)$, $\nabla_x f_{t,i}=\nabla_x f_i(x_t^i,y_t^i)$ and $\nabla_y f_{t,i}=\nabla_y f_i(x_t^i,y_t^i)$.

$$\bar{x}_t=\frac{1}{N}\sum_{i=1}^{N}x_t^i \quad \bar{y}_t=\frac{1}{N}\sum_{i=1}^{N}y_t^i \quad \bar{u}_t=\frac{1}{N}\sum_{i=1}^{N}u_t^i \quad \bar{v}_t=\frac{1}{N}\sum_{i=1}^{N}v_t^i$$

$$\nabla_x\bar{F}_t=\frac{1}{N}\sum_{i=1}^{N}\nabla_x f_{t,i}=\frac{1}{N}\sum_{i=1}^{N}\nabla_x f_i(x_t^i,y_t^i) \quad \nabla_y\bar{F}_t=\frac{1}{N}\sum_{i=1}^{N}\nabla_y f_{t,i}=\frac{1}{N}\sum_{i=1}^{N}\nabla_y f_i(x_t^i,y_t^i)$$

$$\nabla_x F(\bar{x}_t,\bar{y}_t)=\frac{1}{N}\sum_{i=1}^{N}\nabla_x f_i(\bar{x}_t,\bar{y}_t) \quad \nabla_y F(\bar{x}_t,\bar{y}_t)=\frac{1}{N}\sum_{i=1}^{N}\nabla_y f_i(\bar{x}_t,\bar{y}_t)$$

$s_t$ denotes the $s_t=\lfloor t/Q\rfloor$.

### C.1 Important Conclusions

**Lemma C.1.** *Assume sequences $\{\bar{x}_t, \bar{y}_t\}_{t=0}^T$ are generated from Algorithm 2, $i \in [N]$, we have*

$$\mathbb{E}\|\nabla_x f_i(x_t^i, y_t^i; \mathcal{B}_t^i) - \nabla_x f_{t,i}\|^2 \leq \frac{\sigma^2}{b} \tag{25}$$

$$\mathbb{E}\|\nabla_y f_i(x_t^i, y_t^i; \mathcal{B}_t^i) - \nabla_y f_{t,i}\|^2 \leq \frac{\sigma^2}{b} \tag{26}$$

$$\sum_{i=1}^N \mathbb{E}\|\nabla_x f_{t,i} - \nabla_x \bar{F}_t\|^2 \leq 6L_f^2 \sum_{i=1}^N \mathbb{E}[\|x_t^i - \bar{x}_t\|^2 + \|y_t^i - \bar{y}_t\|^2] + 3N\zeta^2 \tag{27}$$

$$\sum_{i=1}^N \mathbb{E}\|\nabla_y f_{t,i} - \nabla_y \bar{F}_t\|^2 \leq 6L_f^2 \sum_{i=1}^N \mathbb{E}[\|x_t^i - \bar{x}_t\|^2 + \|y_t^i - \bar{y}_t\|^2] + 3N\zeta^2 \tag{28}$$

*Proof.* (1) we have

$$\mathbb{E}\|\nabla_x f_i(x_t^i, y_t^i; \mathcal{B}_t^i) - \nabla_x f_{t,i}\|^2 = \mathbb{E}\|\frac{1}{b} \sum_{\xi_{t,i} \in \mathcal{B}_t^i} (\nabla_x f_i(x_t^i, y_t^i; \xi_{t,i}) - \nabla_x f_{t,i})\|^2$$

$$= \frac{1}{b^2} \sum_{\xi_{t,i} \in \mathcal{B}_t^i} \mathbb{E}\|\nabla_x f_i(x_t^i, y_t^i; \xi_{t,i}) - \nabla_x f_{t,i}\|^2 \leq \frac{\sigma^2}{b}$$

where the second equality is due to $\mathbb{E}_{\xi_{t,i}}[\nabla_x f_i(x_t^i, y_t^i; \xi_{t,i}) - \nabla_x f_{t,i}] = 0$ and the last inequality follows Assumptions 3.1. Similarly, we get the (26)

(2)

$$\sum_{i=1}^N \mathbb{E}\|\nabla_x f_{t,i} - \nabla_x \bar{F}_t\|^2$$

$$\leq 3 \sum_{i=1}^N \mathbb{E}\left[\|\nabla_x f_{t,i} - \nabla_x f_i(\bar{x}_t, \bar{y}_t)\|^2 + \|\nabla_x F(\bar{x}_t, \bar{y}_t) - \nabla_x \bar{F}_t\|^2 + \|\nabla_x f_i(\bar{x}_t, \bar{y}_t) - \nabla_x F(\bar{x}_t, \bar{y}_t)\|^2\right]$$

$$\leq 6L_f^2 \sum_{i=1}^N [\mathbb{E}\|x_t^i - \bar{x}_t\|^2 + \mathbb{E}\|y_t^i - \bar{y}_t\|^2] + 3N\zeta^2$$

where the last inequality is due to Assumption 3.3 and 3.1. Similarly, we get the (28) $\qquad\square$

**Lemma C.2.** *Suppose sequences $\{\mathbf{x}_t, \mathbf{y}_t\}_{t=0}^T$ are generated from Algorithms 2. We have*

$$\mathbb{E}\Phi(\bar{x}_{t+1}) \leq \mathbb{E}\Phi(\bar{x}_t) - \left(\frac{\hat{c}\eta}{2} - \frac{\hat{c}^2 \eta^2 L}{2}\right) \mathbb{E}\|\bar{u}_{t+1}\|^2 + \frac{3\hat{c}\eta}{2}\mathbb{E}\|\bar{u}_{t+1} - \frac{1}{N}\sum_{i=1}^N \nabla_x f_i(x_t^i, y_t^i)\|^2$$

$$- \frac{\hat{c}\eta}{2}\mathbb{E}\|\nabla\Phi(\bar{x}_t)\|^2 + \frac{3\hat{c}\eta L_f^2}{2N}\sum_{i=1}^N \mathbb{E}[\|x_t^i - \bar{x}_t\|^2 + \|y_t^i - \bar{y}_t\|^2] + \frac{3\hat{c}\eta L_f^2}{\mu}\mathbb{E}[\Phi(\bar{x}_t) - F(\bar{x}_t, \bar{y}_t)]$$

*Proof.*

$$\Phi(\bar{x}_{t+1}) \leq \Phi(\bar{x}_t) + \langle\nabla\Phi(\bar{x}_t), \bar{x}_{t+1} - \bar{x}_t\rangle + \frac{L}{2}\|\bar{x}_{t+1} - \bar{x}_t\|^2$$

$$= \Phi(\bar{x}_t) + \hat{c}\eta\langle\nabla\Phi(\bar{x}_t), \bar{u}_{t+1}\rangle + \frac{L\hat{c}^2\eta^2}{2}\|\bar{u}_{t+1}\|^2$$

$$= \Phi(\bar{x}_t) - \frac{\hat{c}\eta}{2}\|\bar{u}_{t+1}\|^2 - \frac{\hat{c}\eta}{2}\|\nabla\Phi(\bar{x}_t)\|^2 + \frac{\hat{c}\eta}{2}\|\bar{u}_{t+1} - \nabla\Phi(\bar{x}_t)\|^2 + \frac{\hat{c}^2\eta^2 L}{2}\|\bar{u}_{t+1}\|^2$$

$$\leq \Phi(\bar{x}_t) - \frac{\hat{c}\eta}{2}\|\nabla\Phi(\bar{x}_t)\|^2 - \left(\frac{\hat{c}\eta}{2} - \frac{\hat{c}^2\eta^2 L}{2}\right)\|\bar{u}_{t+1}\|^2 + \frac{3\hat{c}\eta}{2}\|\bar{u}_{t+1} - \frac{1}{N}\sum_{i=1}^N \nabla_x f_i(x_t^i, y_t^i)\|^2$$

$$+ \frac{3\hat{c}\eta}{2}\|\nabla_x F(\bar{x}_t, \bar{y}_t) - \frac{1}{N}\sum_{i=1}^N \nabla_x f_i(x_t^i, y_t^i)\|^2 + \frac{3\hat{c}\eta}{2}\|\nabla\Phi(\bar{x}_t) - \nabla_x F(\bar{x}_t, \bar{y}_t)\|^2$$

Taking expectation on both sides and considering

$$\mathbb{E}\|\nabla_x F(\bar{x}_t, \bar{y}_t) - \nabla_x \bar{F}_t\|^2 \leq \frac{1}{N}\sum_{i=1}^{N}\mathbb{E}\|\nabla_x f_i(\bar{x}_t, \bar{y}_t) - \nabla_x f_{t,i}\|^2$$

$$\leq \frac{L_f^2}{N}\sum_{i=1}^{N}\mathbb{E}\|x_t^i - \bar{x}_t\|^2 + \frac{L_f^2}{N}\sum_{i=1}^{N}\mathbb{E}\|y_t^i - \bar{y}_t\|^2$$

$$\mathbb{E}\|\nabla\Phi(\bar{x}_t) - \nabla_x F(\bar{x}_t, \bar{y}_t)\|^2 \leq L_f^2\mathbb{E}\|y^*(\bar{x}_t) - \bar{y}_t\|^2 \leq \frac{2L_f^2\mathbb{E}[\Phi(\bar{x}_t) - F(\bar{x}_t, \bar{y}_t)]}{\mu}$$

where the last inequality holds due to Lemma A.1 (2). Therefore, we obtain

$$\mathbb{E}\Phi(\bar{x}_{t+1}) \leq \mathbb{E}\Phi(\bar{x}_t) - \left(\frac{\hat{c}\eta}{2} - \frac{\hat{c}^2\eta^2 L}{2}\right)\mathbb{E}\|\bar{u}_{t+1}\|^2 + \frac{3\hat{c}\eta}{2}\mathbb{E}\|\bar{u}_{t+1} - \frac{1}{N}\sum_{i=1}^{N}\nabla_x f_i\left(x_t^i, y_t^i\right)\|^2$$

$$-\frac{\hat{c}\eta}{2}\mathbb{E}\|\nabla\Phi\left(\bar{x}_t\right)\|^2 + \frac{3\hat{c}\eta L_f^2}{2N}\sum_{i=1}^{N}\mathbb{E}[\|x_t^i - \bar{x}_t\|^2 + \|y_t^i - \bar{y}_t\|^2] + \frac{3\hat{c}\eta L_f^2}{\mu}\mathbb{E}[\Phi(\bar{x}_t) - F(\bar{x}_t, \bar{y}_t)]$$

$$\square$$

**Lemma C.3.** *Under the assumptions 3.1, 3.9 and 3.11, and set $\hat{c} \leq \frac{c}{4\kappa^2}$ and $\eta \leq \frac{1}{20QL_f}$, for the Algorithm 2, we have*

$$[\Phi(\bar{x}_{t+1}) - F(\bar{x}_{t+1}, \bar{y}_{t+1})] - [\Phi(\bar{x}_t) - F(\bar{x}_t, \bar{y}_t)]$$

$$\leq -\frac{c\eta\mu}{2}[\Phi(\bar{x}_t) - F(\bar{x}_t, \bar{y}_t)] + \frac{\eta}{4\hat{c}}\|\hat{x}_{t+1} - \bar{x}_t\|^2 - \frac{\eta c}{4}\|\bar{v}_{t+1}\|^2 + \frac{3L_f^2 c\eta}{N}\sum_{i=1}^{N}\|x_t^i - \bar{x}_t\|^2$$

$$+\frac{3L_f^2 c\eta}{N}\sum_{i=1}^{N}\|y_t^i - \bar{y}_t\|^2 + 3c\eta\|\bar{v}_{t+1} - \nabla_y\bar{F}_t\|^2 \tag{29}$$

*where $\kappa = L_{21}/\mu$.*

*Proof.* Define $\hat{x}_{t+1} = \bar{x}_t - \hat{c}\bar{u}_{t+1}$ and $\hat{y}_{t+1} = \bar{y}_t + c\bar{v}_{t+1}$. The function $F(x, y)$ is $L_f$-smooth w.r.t $y$, and we have

$$F(\bar{x}_{t+1}, \bar{y}_t) \leq F(\bar{x}_{t+1}, \bar{y}_{t+1}) - \langle\nabla_y F(\bar{x}_{t+1}, \bar{y}_t), \bar{y}_{t+1} - \bar{y}_t\rangle + \frac{L_f}{2}\|\bar{y}_{t+1} - \bar{y}_t\|^2 \tag{30}$$

For the second term in the (30), we have

$$-\langle\nabla_y F(\bar{x}_{t+1}, \bar{y}_t), \bar{y}_{t+1} - \bar{y}_t\rangle = -c\eta\langle\nabla_y F(\bar{x}_{t+1}, \bar{y}_t), \bar{v}_{t+1}\rangle$$

$$= -\frac{c\eta}{2}\left[\|\nabla_y F(\bar{x}_{t+1}, \bar{y}_t)\|^2 + \|\bar{v}_{t+1}\|^2 - \|\nabla_y F(\bar{x}_{t+1}, \bar{y}_t) - \nabla_y F(\bar{x}_t, \bar{y}_t) + \nabla_y F(\bar{x}_t, \bar{y}_t) - \bar{v}_{t+1}\|^2\right]$$

$$\leq -c\eta\mu[\Phi(\bar{x}_{t+1}) - F(\bar{x}_{t+1}, \bar{y}_t)] - \frac{c\eta}{2}\|\bar{v}_{t+1}\|^2 + c\eta\left[L_f^2\|\bar{x}_{t+1} - \bar{x}_t\|^2 + \|\nabla_y F(\bar{x}_t, \bar{y}_t) - \bar{v}_{t+1}\|^2\right]$$

where the last inequality from the quadratic growth condition of $\mu$-PL functions (9), Combining above two inequalities, we get

$$F(\bar{x}_{t+1}, \bar{y}_t) \leq F(\bar{x}_{t+1}, \bar{y}_{t+1}) - c\eta\mu[\Phi(\bar{x}_{t+1}) - F(\bar{x}_{t+1}, \bar{y}_t)] - \frac{c\eta}{2}\|\bar{v}_{t+1}\|^2$$

$$+ \frac{L_f}{2}\|\bar{y}_{t+1} - \bar{y}_t\|^2 + c\eta\left[L_f^2\|\bar{x}_{t+1} - \bar{x}_t\|^2 + \|\nabla_y F(\bar{x}_t, \bar{y}_t) - \bar{v}_{t+1}\|^2\right]$$

Rearranging the above inequality, we get

$$\Phi(\bar{x}_{t+1}) - F(\bar{x}_{t+1}, \bar{y}_{t+1}) \leq (1 - c\eta\mu)[\Phi(\bar{x}_{t+1}) - F(\bar{x}_{t+1}, \bar{y}_t)] - \frac{c\eta - L_f c^2\eta^2}{2}\|\bar{v}_{t+1}\|^2$$

$$+ c\eta\left[L_f^2\|\bar{x}_{t+1} - \bar{x}_t\|^2 + \|\nabla_y F(\bar{x}_t, \bar{y}_t) - \bar{v}_{t+1}\|^2\right] \tag{31}$$

Furthermore, we bound $\Phi\left(\bar{x}_{t+1}\right) - F\left(\bar{x}_{t+1}, \bar{y}_t\right)$

$$\begin{aligned}
&\Phi\left(\bar{x}_{t+1}\right) - F\left(\bar{x}_{t+1}, \bar{y}_t\right) \\
&= \Phi\left(\bar{x}_{t+1}\right) - \Phi\left(\bar{x}_t\right) + \left[\Phi\left(\bar{x}_t\right) - F\left(\bar{x}_t, \bar{y}_t\right)\right] + F\left(\bar{x}_t, \bar{y}_t\right) - F\left(\bar{x}_{t+1}, \bar{y}_t\right).
\end{aligned} \tag{32}$$

Considering the last two terms, we have

$$\begin{aligned}
&F\left(\bar{x}_t, \bar{y}_t\right) - F\left(\bar{x}_{t+1}, \bar{y}_t\right) \\
&\overset{(a)}{\leq} -\left\langle \nabla_x F\left(\bar{x}_t, \bar{y}_t\right), \bar{x}_{t+1} - \bar{x}_t \right\rangle + \frac{L_f}{2}\left\|\bar{x}_{t+1} - \bar{x}_t\right\|^2 \\
&\overset{(b)}{=} -\eta\left\langle \nabla_x F\left(\bar{x}_t, \bar{y}_t\right) - \nabla\Phi\left(\bar{x}_t\right), \hat{x}_{t+1} - \bar{x}_t \right\rangle - \eta\left\langle \nabla\Phi\left(\bar{x}_t\right), \hat{x}_{t+1} - \bar{x}_t \right\rangle + \frac{L_f}{2}\left\|\bar{x}_{t+1} - \bar{x}_t\right\|^2 \\
&\overset{(c)}{\leq} \frac{\eta}{8\hat{c}}\left\|\hat{x}_{t+1} - \bar{x}_t\right\|^2 + 2\hat{c}\eta\left\|\nabla_x F\left(\bar{x}_t, \bar{y}_t\right) - \nabla\Phi\left(\bar{x}_t\right)\right\|^2 + \Phi\left(\bar{x}_t\right) - \Phi\left(\bar{x}_{t+1}\right) + \frac{L + L_f}{2}\left\|\bar{x}_{t+1} - \bar{x}_t\right\|^2 \\
&\overset{(d)}{\leq} \frac{\eta}{8\hat{c}}\left\|\hat{x}_{t+1} - \bar{x}_t\right\|^2 + \frac{4\hat{c}\eta L_f^2}{\mu}\left[\Phi\left(\bar{x}_t\right) - F\left(\bar{x}_t, \bar{y}_t\right)\right] + \Phi\left(\bar{x}_t\right) - \Phi\left(\bar{x}_{t+1}\right) + \frac{L + L_f}{2}\left\|\bar{x}_{t+1} - \bar{x}_t\right\|^2 \\
&\leq \frac{4\hat{c}\eta L_f^2}{\mu}\left[\Phi\left(\bar{x}_t\right) - F\left(\bar{x}_t, \bar{y}_t\right)\right] + \Phi\left(\bar{x}_t\right) - \Phi\left(\bar{x}_{t+1}\right) + L\left\|\bar{x}_{t+1} - \bar{x}_t\right\|^2 + \frac{\eta}{8\hat{c}}\left\|\hat{x}_{t+1} - \bar{x}_t\right\|^2 \\
&= \frac{4\hat{c}\eta L_f^2}{\mu}\left[\Phi\left(\bar{x}_t\right) - F\left(\bar{x}_t, \bar{y}_t\right)\right] + \Phi\left(\bar{x}_t\right) - \Phi\left(\bar{x}_{t+1}\right) + [\eta^2 L + \frac{\eta}{8\hat{c}}]\left\|\hat{x}_{t+1} - \bar{x}_t\right\|^2
\end{aligned} \tag{33}$$

where (a) holds due to $L_f$-smoothness of $F(\cdot, y_t)$; (b) holds due to the definition of $\hat{x}_{t+1}$ and $\bar{x}_{t+1} - \bar{x}_t = \eta(\hat{x}_{t+1} - \bar{x}_t)$; (c) follows from Young inequality and smoothness of $\Phi(\cdot)$; (d) follows by the property of PL condition (8). Combining (31), (32) and (33), we have

$$\begin{aligned}
&\Phi\left(\bar{x}_{t+1}\right) - F\left(\bar{x}_{t+1}, \bar{y}_{t+1}\right) \\
&\leq (1 - c\eta\mu)\left[\left(1 + \frac{4\hat{c}\eta L_f^2}{\mu}\right)\left[\Phi\left(\bar{x}_t\right) - F\left(\bar{x}_t, \bar{y}_t\right)\right] + [\eta^2 L + \frac{\eta}{8\hat{c}}]\left\|\hat{x}_{t+1} - \bar{x}_t\right\|^2\right] \\
&\quad - \frac{c\eta - L_f c^2\eta^2}{2}\left\|\bar{v}_{t+1}\right\|^2 + c\eta\left[L_f^2\left\|\bar{x}_{t+1} - \bar{x}_t\right\|^2 + \left\|\nabla_y F\left(\bar{x}_t, \bar{y}_t\right) - \bar{v}_{t+1}\right\|^2\right] \\
&\overset{(a)}{\leq} \left(1 - \frac{c\eta\mu}{2}\right)\left[\Phi\left(\bar{x}_t\right) - F\left(\bar{x}_t, \bar{y}_t\right)\right] + [\eta^2 L + \frac{\eta}{8\hat{c}} + \eta^3 L_f^2 c]\left\|\hat{x}_{t+1} - \bar{x}_t\right\|^2 \\
&\quad - \frac{c\eta - L_f c^2\eta^2}{2}\left\|\bar{v}_{t+1}\right\|^2 + c\eta\left\|\nabla_y F\left(\bar{x}_t, \bar{y}_t\right) - \bar{v}_{t+1}\right\|^2 \\
&\overset{(b)}{\leq} \left(1 - \frac{c\eta\mu}{2}\right)\left[\Phi\left(\bar{x}_t\right) - F\left(\bar{x}_t, \bar{y}_t\right)\right] + \frac{\eta}{4\hat{c}}\left\|\hat{x}_{t+1} - \bar{x}_t\right\|^2 - \frac{\eta c}{4}\left\|\bar{v}_{t+1}\right\|^2 \\
&\quad + c\eta\left\|\nabla_y F\left(\bar{x}_t, \bar{y}_t\right) - \bar{v}_{t+1}\right\|^2
\end{aligned} \tag{34}$$

where in $(a)$ we need $(1 - c\eta\mu)\left(1 + \frac{4\hat{c}\eta L_f^2}{\mu}\right) \leq \left(1 - \frac{c\eta\mu}{2}\right)$. This holds if $\frac{4\hat{c}\eta L_f^2}{\mu} \leq \frac{c\eta\mu}{2} \Rightarrow \hat{c} \leq \frac{c}{8\kappa^2}$; (b) follows since $\eta \leq \frac{1}{20\hat{c}L_f}$ and $c\eta \leq \frac{1}{20L_f}$. Rearranging the term, we have

$$\begin{aligned}
&\left[\Phi\left(\bar{x}_{t+1}\right) - F\left(\bar{x}_{t+1}, \bar{y}_{t+1}\right)\right] - \left[\Phi\left(\bar{x}_t\right) - F\left(\bar{x}_t, \bar{y}_t\right)\right] \\
&\leq -\frac{c\eta\mu}{2}\left[\Phi\left(\bar{x}_t\right) - F\left(\bar{x}_t, \bar{y}_t\right)\right] + \frac{\eta}{4\hat{c}}\left\|\hat{x}_{t+1} - \bar{x}_t\right\|^2 - \frac{\eta c}{4}\left\|\bar{v}_{t+1}\right\|^2 + c\eta\left\|\nabla_y F\left(\bar{x}_t, \bar{y}_t\right) - \bar{v}_{t+1}\right\|^2 \\
&\leq -\frac{c\eta\mu}{2}\left[\Phi\left(\bar{x}_t\right) - F\left(\bar{x}_t, \bar{y}_t\right)\right] + \frac{\eta}{4\hat{c}}\left\|\hat{x}_{t+1} - \bar{x}_t\right\|^2 - \frac{\eta c}{4}\left\|\bar{v}_{t+1}\right\|^2 + \frac{3L_f^2 c\eta}{N}\left\|x_t - \bar{x}_t\right\|^2 \\
&\quad + \frac{3L_f^2 c\eta}{N}\left\|y_t - \bar{y}_t\right\|^2 + 3c\eta\left\|\bar{v}_{t+1} - \nabla_y \bar{F}_t\right\|^2
\end{aligned} \tag{35}$$

where $\left\|\nabla_y F\left(\bar{x}_t, \bar{y}_t\right) - \bar{v}_{t+1}\right\|^2 \leq \frac{3L_f^2}{N}\sum_{i=1}^{N}\left\|x_t^i - \bar{x}_t\right\|^2 + \frac{3L_f^2}{N}\sum_{i=1}^{N}\left\|y_t^i - \bar{y}_t\right\|^2 + 3\left\|\bar{v}_{t+1} - \nabla_y \bar{F}_t\right\|^2$

$\square$

**Lemma C.4.** *For every $t \in [0, T]$ the iterates generated by Algorithm 2 satisfy*

$$\mathbb{E}\|\bar{u}_{t+1} - \nabla_x \bar{F}_t\|^2 = (1-\alpha)^2 \mathbb{E}\|\bar{u}_t - \nabla_x \bar{F}_{t-1}\|^2 + \frac{8(1-\alpha)^2 L_f^2}{N^2 b} \frac{Q-1}{Q} \sum_{i=1}^{N} \mathbb{E}[\hat{c}^2 \eta^2 \|u_t^i - \bar{u}_t\|^2$$

$$+ c^2 \eta^2 \|v_t^i - \bar{v}_t\|^2] + \frac{4(1-\alpha)^2 L_f^2}{Nb} \mathbb{E}[\hat{c}^2 \eta^2 \|\bar{u}_t\|^2 + c^2 \eta^2 \|\bar{v}_t\|^2] + \frac{2\alpha^2 \sigma^2}{Nb}$$

$$\mathbb{E}\|\bar{v}_{t+1} - \nabla_y \bar{F}_t\|^2 = (1-\beta)^2 \mathbb{E}\|\bar{v}_t - \nabla_y \bar{F}_{t-1}\|^2 + \frac{8(1-\beta)^2 L_f^2}{N^2 b} \frac{Q-1}{Q} \sum_{i=1}^{N} \mathbb{E}[\hat{c}^2 \eta^2 \|u_t^i - \bar{u}_t\|^2$$

$$+ c^2 \eta^2 \|v_t^i - \bar{v}_t\|^2] + \frac{4(1-\beta)^2 L_f^2}{Nb} \mathbb{E}[\hat{c}^2 \eta^2 \|\bar{u}_t\|^2 + c^2 \eta^2 \|\bar{v}_t\|^2] + \frac{2\beta^2 \sigma^2}{Nb}$$

*Proof.* We borrow the proof steps from Lemma A.6 [34], which study the application of momentum-based variance reduced technology in FL. Recall the definition of momentum-based variance reduced estimator $\bar{u}_{t+1} = \frac{1}{N} \sum_{i=1}^{N} [\nabla_x f_i(x_t^i, y_t^i; \mathcal{B}_t^i) + (1-\alpha)(\bar{u}_t - \nabla_x f_i(x_{t-1}^i, y_{t-1}^i; \mathcal{B}_t^i))]$ and $\bar{v}_{t+1} = \frac{1}{N} \sum_{i=1}^{N} [\nabla_y f_i(x_t^i, y_t^i; \mathcal{B}_t^i) + (1-\beta)(\bar{u}_t - \nabla_y f_i(x_{t-1}^i, y_{t-1}^i; \mathcal{B}_t^i))]$, we have easily get the final results. $\square$

**Lemma C.5.** *Assume that the stochastic partial derivatives $\{\bar{u}_t\}$ and $\{\bar{v}_t\}$ are generated from Algorithm 2, and set $s_t = \lfloor t/Q \rfloor$, $\eta \leq \frac{1}{20QL}$, $\alpha = c_1 \eta^2$, $\beta = c_2 \eta^2$, where we set $c_1 \leq \frac{30L^2}{bN}$ and $c_2 \leq \frac{30L^2}{bN}$, we have*

$$\frac{2}{5N} \sum_{t=s_t Q}^{\bar{s}} \sum_{i=1}^{N} \mathbb{E}[\|u_t^i - \bar{u}_t\|^2 + \|v_t^i - \bar{v}_t\|^2] \leq \frac{1}{12} \sum_{t=s_t Q}^{\bar{s}} \mathbb{E}[\hat{c}^2 \|\bar{u}_t\|^2 + c^2 \|\bar{v}_t\|^2]$$

$$+ \left[ \frac{\sigma^2(c_1^2 + c_2^2)}{30bL^2} + \frac{\zeta^2(c_1^2 + c_2^2)}{12L^2} \right] \sum_{t=s_t Q}^{\bar{s}} \eta^2 \tag{36}$$

*Proof.*

$$\sum_{i=1}^{N} \mathbb{E}\|u_{t+1}^i - \bar{u}_{t+1}\|^2$$

$$= \sum_{i=1}^{N} \mathbb{E}\|\nabla_x f_i(x_t^i, y_t^i; \mathcal{B}_t^i) + (1-\alpha)(u_t^i - \nabla_x f_i(x_{t-1}^i, y_{t-1}^i; \mathcal{B}_t^i))$$

$$- \frac{1}{N} \sum_{j=1}^{N} [\nabla_x f_j(x_t^j, y_t^j; \mathcal{B}_t^j) + (1-\alpha)(u_t^j - \nabla_x f_j(x_{t-1}^j, y_{t-1}^j; \mathcal{B}_t^j))]\|^2$$

$$= \sum_{i=1}^{N} \mathbb{E}\|(1-\alpha)(u_t^i - \bar{u}_t) + [\nabla_x f_i(x_t^i, y_t^i; \mathcal{B}_t^i) - \frac{1}{N} \sum_{i=1}^{N} \nabla_x f_j(x_t^j, y_t^j; \mathcal{B}_t^j)$$

$$- (1-\alpha)[\nabla_x f_i(x_{t-1}^i, y_{t-1}^i; \mathcal{B}_t^i) - \frac{1}{N} \sum_{j=1}^{N} \nabla_x f_j(x_{t-1}^j, y_{t-1}^j; \mathcal{B}_t^j)]\|^2$$

$$\overset{(a)}{\leq} (1+\gamma)(1-\alpha)^2 \sum_{i=1}^{N} \mathbb{E}\|u_t^i - \bar{u}_t\|^2$$

$$+ (1+\frac{1}{\gamma}) \sum_{i=1}^{N} \mathbb{E}\|[\nabla_x f_i(x_t^i, u_t^i; \mathcal{B}_t^i) - \frac{1}{N} \sum_{j=1}^{N} \nabla_x f_j(x_t^j, y_t^j; \mathcal{B}_t^j)]$$

$$- (1-\alpha)[\nabla_x f_i(x_{t-1}^i, y_{t-1}^i; \mathcal{B}_t^i) - \frac{1}{N} \sum_{j=1}^{N} \nabla_x f_j(x_{t-1}^j, y_{t-1}^j; \mathcal{B}_t^j)]\|^2$$

where (a) is due to Young's inequality. For the second term, we have

$$\sum_{i=1}^{N} \mathbb{E}\|\nabla_x f_i(x_t^i, y_t^i; \mathcal{B}_t^i) - \frac{1}{N}\sum_{j=1}^{N} \nabla_x f_j(x_t^j, y_t^j; \mathcal{B}_t^j)$$

$$- (1-\alpha)[\nabla_x f_i(x_{t-1}^i, y_{t-1}^i; \mathcal{B}_t^i) - \frac{1}{N}\sum_{j=1}^{N} \nabla_x f_j(x_{t-1}^j, y_{t-1}^j; \mathcal{B}_t^j)]\|^2$$

$$\leq 2\sum_{i=1}^{N} \mathbb{E}\|[\nabla_x f_i(x_t^i, y_t^i; \mathcal{B}_t^i) - \frac{1}{N}\sum_{j=1}^{N} \nabla_x f_j(x_t^j, y_t^j; \mathcal{B}_t^j)]$$

$$- [\nabla_x f_i(x_{t-1}^i, y_{t-1}^i; \mathcal{B}_t^i) - \frac{1}{N}\sum_{j=1}^{N} \nabla_x f_j(x_{t-1}^j, y_{t-1}^j; \mathcal{B}_t^j)]\|^2$$

$$+ 2\alpha^2 \sum_{i=1}^{N} \mathbb{E}\|\nabla_x f_i(x_{t-1}^i, y_{t-1}^i; \mathcal{B}_t^i) - \frac{1}{N}\sum_{j=1}^{N} \nabla_x f_j(x_{t-1}^j, y_{t-1}^j; \mathcal{B}_t^j)\|^2$$

$$\overset{(a)}{\leq} 2\sum_{i=1}^{N} \mathbb{E}\|\nabla_x f_i(x_t^i, y_t^i; \mathcal{B}_t^i) - \nabla_x f_i(x_{t-1}^i, y_{t-1}^i; \mathcal{B}_t^i)\|^2$$

$$+ 2\alpha^2 \sum_{i=1}^{N} \mathbb{E}\|\nabla_x f_i(x_{t-1}^i, y_{t-1}^i; \mathcal{B}_t^i) - \frac{1}{N}\sum_{j=1}^{N} \nabla_x f_j(x_{t-1}^j, y_{t-1}^j; \mathcal{B}_t^j)\|^2$$

$$\overset{(b)}{\leq} 2L_f^2 \sum_{i=1}^{N} \mathbb{E}[\|x_t^i - x_{t-1}^i\|^2 + \|y_t^i - y_{t-1}^i\|^2] + 2\alpha^2 \sum_{i=1}^{N} \mathbb{E}\|\nabla_x f_i(x_{t-1}^i, y_{t-1}^i; \mathcal{B}_t^i)$$

$$- \frac{1}{N}\sum_{j=1}^{N} \nabla_x f_j(x_{t-1}^j, y_{t-1}^j; \mathcal{B}_t^j)\|^2$$

where (a) is due to Lemma A.2; (b) is due to Assumption 3.9. For the last term in the above inequality, we have

$$\sum_{i=1}^{N} \mathbb{E}\|\nabla_x f_i(x_{t-1}^i, y_{t-1}^i; \mathcal{B}_t^i) - \frac{1}{N}\sum_{j=1}^{N} \nabla_x f_j(x_{t-1}^j, y_{t-1}^j; \mathcal{B}_t^j)\|^2$$

$$= \sum_{i=1}^{N} \mathbb{E}\|[\nabla_x f_i(x_{t-1}^i, y_{t-1}^i; \mathcal{B}_t^i) - \nabla_x f_{t-1,i}] - \frac{1}{N}\sum_{j=1}^{N} [\nabla_x f_j(x_{t-1}^j, y_{t-1}^j; \mathcal{B}_t^j) - \nabla_x f_{t-1,j}]$$

$$+ [\nabla_x f_{t-1,i} - \nabla_x \bar{F}_{t-1}]\|^2$$

$$\leq 2\sum_{i=1}^{N} \mathbb{E}\|[\nabla_x f_i(x_{t-1}^i, y_{t-1}^i; \mathcal{B}_t^i) - \nabla_x f_{t-1,i}] - \frac{1}{N}\sum_{j=1}^{N} [\nabla_x f_j(x_{t-1}^j, y_{t-1}^j; \mathcal{B}_t^j) - \nabla_x f_{t-1,j}]\|^2$$

$$+ 2\sum_{i=1}^{N} \mathbb{E}\|\nabla_x f_{t-1,i} - \nabla_x \bar{F}_{t-1}\|^2$$

$$\overset{(a)}{\leq} 2\sum_{i=1}^{N} \mathbb{E}\|\nabla_x f_i(x_{t-1}^i, y_{t-1}^i; \mathcal{B}_t^i) - \nabla_x f_{t-1,i}\|^2 + 2\sum_{i=1}^{N} \mathbb{E}\|\nabla_x f_{t-1,i} - \nabla_x \bar{F}_{t-1}\|^2$$

$$\overset{(b)}{\leq} \frac{2N\sigma^2}{b} + 6N\zeta^2 + 12L_f^2 \sum_{i=1}^{N} \mathbb{E}\|x_{t-1}^i - \bar{x}_{t-1}\|^2 + 12L_f^2 \sum_{i=1}^{N} \mathbb{E}\|y_{t-1}^i - \bar{y}_{t-1}\|^2 \qquad (37)$$

where (a) is due to Lemma A.2 and the last inequality is due to Lemma C.1. Therefore, by combining above inequalities, we have

$$
\sum_{i=1}^{N} \mathbb{E}\|u_{t+1}^i - \bar{u}_{t+1}\|^2 \leq (1-\alpha)^2(1+\gamma) \sum_{i=1}^{N} \mathbb{E}\|u_t^i - \bar{u}_t\|^2 + \frac{4N\sigma^2}{b}(1+\frac{1}{\gamma})\alpha^2
$$

$$
+ 12N\zeta^2(1+\frac{1}{\gamma})\alpha^2 + 2L_f^2(1+\frac{1}{\gamma}) \sum_{i=1}^{N} \mathbb{E}[\|x_{t,i} - x_{t-1}^i\|^2 + \|y_t^i - y_{t-1}^i\|^2]
$$

$$
+ 24L_f^2(1+\frac{1}{\gamma})\alpha^2 \sum_{i=1}^{N} \mathbb{E}[\|x_{t-1}^i - \bar{x}_{t-1,i}\|^2 + \|y_{t-1}^i - \bar{y}_{t-1,i}\|^2]
$$

$$
\leq (1-\alpha)^2(1+\gamma) \sum_{i=1}^{N} \mathbb{E}\|u_t^i - \bar{u}_t\|^2 + \frac{4N\sigma^2}{b}(1+\frac{1}{\gamma})\alpha^2 + 12N\zeta^2(1+\frac{1}{\gamma})\alpha^2
$$

$$
+ 4L_f^2(1+\frac{1}{\gamma}) \sum_{i=1}^{N} \mathbb{E}\big[\|\hat{c}\eta(u_t^i - \bar{u}_t)\|^2 + \|\hat{c}\eta\bar{u}_t\|^2 + \|c\eta(v_t^i - \bar{v}_t)\|^2 + \|c\eta\bar{v}_t\|^2\big]
$$

$$
+ 24L_f^2(1+\frac{1}{\gamma})\alpha^2(Q-1) \sum_{s=s_t+1}^{t-1} \eta^2 \sum_{i=1}^{N} \mathbb{E}[\|\hat{c}(u_s^i - \bar{u}_s)\|^2 + \|c(v_s^i - \bar{v}_s)\|^2] \tag{38}
$$

where the inequality is due to the update step 11 in algorithm 2 and the fact that. (1) if $t = s_t Q$, we have

$$
\sum_{i=1}^{N} \left\|x_{s_t Q}^i - \bar{x}_{s_t Q}\right\|^2 = 0 \tag{39}
$$

(2) if $t \geq s_t Q$, we have

$$
\sum_{i=1}^{N} \left\|x_t^i - \bar{x}_t\right\|^2 = \sum_{i=1}^{N} \left\|x_{s_t Q}^i - \bar{x}_{s_t Q} - \left( \sum_{s=s_t Q}^{t-1} \hat{c}\eta u_{s+1}^i - \sum_{s=s_t Q}^{t-1} \hat{c}\eta\bar{u}_{s+1} \right)\right\|^2
$$

$$
= \sum_{i=1}^{N} \left\| \sum_{s=s_t Q}^{t-1} \hat{c}\eta \left[ u_{s+1}^i - \bar{u}_{s+1} \right] \right\|^2
$$

$$
\leq (Q-1) \sum_{s=s_t Q}^{t-1} \hat{c}^2\eta^2 \sum_{i=1}^{N} \|u_{s+1}^i - \bar{u}_{s+1}\|^2 \tag{40}
$$

Given that $\alpha, \beta, \hat{c}, c \leq 1$, then based on (38), we have

$$
\sum_{i=1}^{N} \mathbb{E}[\|u_{t+1}^i - \bar{u}_{t+1}\|^2 + \|v_{t+1}^i - \bar{v}_{t+1}\|^2]
$$

$$
= [(1+\gamma) + 8L_f^2(1+\frac{1}{\gamma})\eta^2] \sum_{i=1}^{N} \mathbb{E}[\|u_t^i - \bar{u}_t\|^2 + \|v_t^i - \bar{v}_t\|^2] + 12N\zeta^2(1+\frac{1}{\gamma})(\alpha^2 + \beta^2)
$$

$$
+ 8NL_f^2(1+\frac{1}{\gamma})\eta^2\mathbb{E}[\hat{c}^2\|\bar{u}_t\|^2 + c^2\|\bar{v}_t\|^2] + \frac{4N\sigma^2}{b}(1+\frac{1}{\gamma})(\alpha^2 + \beta^2)
$$

$$
+ 24L_f^2(1+\frac{1}{\gamma})(\alpha^2 + \beta^2)(Q-1) \sum_{s=s_t}^{t-1} \eta^2 \sum_{i=1}^{N} \mathbb{E}[\hat{c}^2\|u_s^i - \bar{u}_s\|^2 + c^2\|v_s^i - \bar{v}_s\|^2] \tag{41}
$$

Set $\gamma = \frac{1}{Q}$ and $\eta \leq \frac{1}{20LQ}$

$$(1+\gamma) + 8L_f^2(1+\frac{1}{\gamma})\eta^2 \leq 1 + \frac{1}{Q} + 8L^2(1+Q)\eta^2$$

$$\leq 1 + \frac{1}{Q} + \frac{Q+1}{50Q^2}$$

$$\leq 1 + \frac{27}{25Q} \tag{42}$$

Putting the Equation (42) in Equation (41), and considering $\gamma = \frac{1}{Q}$ and $c\eta \leq \frac{1}{20LQ}$, we have

$$\sum_{i=1}^{N} \mathbb{E}[\|u_{t+1}^i - \bar{u}_{t+1}\|^2 + \|v_{t+1}^i - \bar{v}_t\|^2]$$

$$\leq (1+\frac{27}{25Q})\sum_{i=1}^{N}\mathbb{E}[\|u_t^i - \bar{u}_t\|^2 + \|v_t^i - \bar{v}_t\|^2] + 8NL_f^2(1+Q)\eta^2\mathbb{E}[\hat{c}^2\|\bar{u}_t\|^2 + c^2\|\bar{v}_t\|^2]$$

$$+ \frac{4N\sigma^2}{b}(1+Q)(\alpha^2+\beta^2) + 12N\zeta^2(1+Q)(\alpha^2+\beta^2)$$

$$+ 24(\alpha^2+\beta^2)L_f^2(1+Q)(Q-1)\sum_{s=s_tQ}^{t-1}\eta^2\sum_{i=1}^{N}\mathbb{E}[\hat{c}^2\|u_s^i - \bar{u}_s\|^2 + c^2\|v_s^i - \bar{v}_s\|^2]$$

$$\leq (1+\frac{27}{25Q})\sum_{i=1}^{N}\mathbb{E}[\|u_t^i - \bar{u}_t\|^2 + \|v_{t,i} - \bar{v}_t\|^2] + \frac{4NL\eta}{5}\mathbb{E}[\hat{c}^2\|\bar{u}_t\|^2 + c^2\|\bar{v}_t\|^2]$$

$$+ \frac{2N\sigma^2(c_1^2+c_2^2)}{5bL}\eta^3 + \frac{6N\zeta^2(c_1^2+c_2^2)}{5L}\eta^3$$

$$+ 24L_f^2Q^2(c_1^2+c_2^2)\eta^4\sum_{s=s_tQ}^{t-1}\eta^2\sum_{i=1}^{N}\mathbb{E}[\hat{c}^2\|u_s^i - \bar{u}_s\|^2 + c^2\|v_s^i - \bar{v}_s\|^2] \tag{43}$$

When $\mod(t,Q) = 0$ (i.e. $t = s_tQ$), $\sum_{i=1}^{N}\|u_t^i - \bar{u}_t\|^2 = 0$ and $\sum_{i=1}^{N}\|v_t^i - \bar{v}_t\|^2 = 0$. Applying (43) recursively , we get

$$\sum_{i=1}^{N} \mathbb{E}[\|u_{t+1}^i - \bar{u}_{t+1}\|^2 + \|v_{t+1}^i - \bar{v}_{t+1}\|^2]$$

$$\leq \frac{4NL}{5}\sum_{s=s_tQ}^{t}(1+\frac{27}{25q})^{t-s}\eta\mathbb{E}[\|\hat{c}\bar{u}_s\|^2 + \|c\bar{v}_s\|^2] + \frac{(2N\sigma^2 + 6N\zeta^2b)(c_1^2+c_2^2)}{5bL}\sum_{s=s_tQ}^{t}(1+\frac{27}{25Q})^{t-s}\eta^3$$

$$+ 24L_f^2Q^2(c_1^2+c_2^2)\sum_{s=s_tQ}^{t}(1+\frac{27}{25Q})^{t-s}\eta^4\sum_{\bar{s}=s_tQ}^{s}\eta_{\bar{s}}^2\sum_{i=1}^{N}\mathbb{E}[\hat{c}^2\|(u_{\bar{s},i} - \bar{u}_{\bar{s}})\|^2 + c^2\|(v_{\bar{s},i} - \bar{v}_{\bar{s}})\|^2]$$

$$\leq \frac{4NL}{5}\sum_{s=s_tQ}^{t}(1+\frac{27}{25Q})^Q\eta\mathbb{E}[\|\hat{c}\bar{u}_s\|^2 + \|c\bar{v}_s\|^2] + \frac{(2N\sigma^2 + 6N\zeta^2b)(c_1^2+c_2^2)}{5bL}\sum_{s=s_tQ}^{t}\left(1+\frac{27}{25Q}\right)^Q\eta^3$$

$$+ 24L_f^2Q^3(c_1^2+c_2^2)(\frac{1}{20LQ})^5(1+\frac{27}{25Q})^Q\sum_{s=s_tQ}^{t}\eta\sum_{i=1}^{N}\mathbb{E}[\hat{c}^2\|u_s^i - \bar{u}_s\|^2 + c^2\|v_s^i - \bar{v}_s\|^2]$$

$$\leq 3NL\sum_{s=s_tQ}^{t}\eta\mathbb{E}[\hat{c}^2\|\bar{u}_s\|^2 + c^2\|\bar{v}_s\|^2] + \frac{(6N\sigma^2 + 18N\zeta^2b)(c_1^2+c_2^2)}{5bL}\sum_{s=s_tQ}^{t}\eta^3$$

$$+ 72L_f^2Q^3(c_1^2+c_2^2)(\frac{1}{20LQ})^5\sum_{s=s_tQ}^{t}\eta\sum_{i=1}^{N}\mathbb{E}[\hat{c}^2\|u_s^i - \bar{u}_s\|^2 + c^2\|v_s^i - \bar{v}_s\|^2] \tag{44}$$

where the third inequality is due to $(1 + 27/25q)^Q \le e^{27/25} \le 3$. Multiplying $\eta$ on both side and summing over $[s_t Q, \bar{s}]$ in one inner loop, we have

$$\sum_{t=s_t Q}^{\bar{s}} \eta \sum_{i=1}^{N} \mathbb{E}[\|u_{t+1}^i - \bar{u}_{t+1}\|^2 + \|v_{t+1}^i - \bar{v}_{t+1}\|^2]$$

$$\le 3NL \sum_{t=s_t Q}^{\bar{s}} \eta \sum_{s=s_t Q}^{t} \eta \mathbb{E}[\|\hat{c}\bar{u}_s\|^2 + \|c\bar{v}_s\|^2] + \frac{(6N\sigma^2 + 18N\zeta^2 b)(c_1^2 + c_2^2)}{5bL} \sum_{t=s_t Q}^{\bar{s}} \eta \sum_{s=s_t Q}^{t} \eta^3$$

$$+ 72L_f^2 Q^3 (c_1^2 + c_2^2)(\frac{1}{20LQ})^5 \sum_{t=s_t Q}^{\bar{s}} \eta \sum_{s=s_t Q}^{t} \eta \sum_{i=1}^{N} \mathbb{E}[\|u_s^i - \bar{u}_s\|^2 + \|v_s^i - \bar{v}_s\|^2]$$

$$\le 3NL (\sum_{t=s_t Q}^{\bar{s}} \eta) \sum_{t=s_t Q}^{\bar{s}} \eta \mathbb{E}[\hat{c}^2 \|\bar{u}_t\|^2 + c^2 \|\bar{v}_t\|^2] + \frac{(6N\sigma^2 + 18N\zeta^2 b)(c_1^2 + c_2^2)}{5bL} (\sum_{t=s_t Q}^{\bar{s}} \eta) \sum_{t=s_t Q}^{\bar{s}} \eta^3$$

$$+ 72L_f^2 Q^3 (c_1^2 + c_2^2)(\frac{1}{20LQ})^5 (\sum_{t=s_t Q}^{\bar{s}} \eta) \sum_{t=s_t Q}^{\bar{s}} \eta \sum_{i=1}^{N} \mathbb{E}[\hat{c}^2 \|u_t^i - \bar{u}_t\|^2 + c^2 \|v_t^i - \bar{v}_t\|^2]$$

$$\le \frac{N}{5} \sum_{t=s_t Q}^{\bar{s}} \eta \mathbb{E}[\hat{c}^2 \|\bar{u}_t\|^2 + c^2 \|\bar{v}_t\|^2] + \left[ \frac{2N\sigma^2(c_1^2 + c_2^2)}{25bL^2} + \frac{N\zeta^2(c_1^2 + c_2^2)}{5L^2} \right] \sum_{t=s_t Q}^{\bar{s}} \eta^3$$

$$+ \frac{72L_f^2 q^4 (c_1^2 + c_2^2)}{(20Lq)^6} \sum_{t=s_t Q}^{\bar{s}} \eta \sum_{i=1}^{N} \mathbb{E}[\|u_t^i - \bar{u}_t\|^2 + \|v_t^i - \bar{v}_t\|^2] \tag{45}$$

where the last inequality holds by the fact that $\eta \le \frac{1}{20LQ}$. Therefore,

$$[1 - 72L^2 q^4 (c_1^2 + c_2^2)(\frac{1}{20LQ})^6] \sum_{t=s_t Q}^{\bar{s}} \eta \sum_{i=1}^{N} \mathbb{E}[\|u_t^i - \bar{u}_t\|^2 + \|v_t^i - \bar{v}_t\|^2]$$

$$\le \frac{N}{5} \sum_{t=s_t Q}^{\bar{s}} \eta \mathbb{E}[\hat{c}^2 \|\bar{u}_t\|^2 + c^2 \|\bar{v}_t\|^2] + \left[ \frac{2N\sigma^2(c_1^2 + c_2^2)}{25bL^2} + \frac{N\zeta^2(c_1^2 + c_2^2)}{5L^2} \right] \sum_{t=s_t Q}^{\bar{s}} \eta^3 \tag{46}$$

Given that $c_1 \le \frac{30L^2}{bN}$ and $c_2 \le \frac{30L^2}{bN}$, and $1 - 72L^2 q^4 (c_1^2 + c_2^2)(\frac{1}{20LQ})^6 \ge \frac{24}{25}$. By multiply $\frac{5}{12N}$ on both size, we have

$$\frac{2}{5N} \sum_{t=s_t Q}^{\bar{s}} \sum_{i=1}^{N} \mathbb{E}[\|u_t^i - \bar{u}_t\|^2 + \|v_t^i - \bar{v}_t\|^2] \le \frac{1}{12} \sum_{t=s_t Q}^{\bar{s}} \mathbb{E}[\hat{c}^2 \|\bar{u}_t\|^2 + c^2 \|\bar{v}_t\|^2]$$

$$+ \left[ \frac{\sigma^2(c_1^2 + c_2^2)}{30bL^2} + \frac{\zeta^2(c_1^2 + c_2^2)}{12L^2} \right] \sum_{t=s_t Q}^{\bar{s}} \eta^2 \tag{47}$$

$$\square$$

## C.2 Proof of Theorem

In this section, we show the Proof of Theorem 3.12.

*Proof.* Set $\eta = \frac{1}{20QL}, \alpha = c_1 \eta^2, \beta = c_2 \eta^2, c_1 = \frac{30L^2}{bN\kappa^{1-\nu}}, c_2 = \frac{30L^2}{bN\kappa^{2-2\nu}}, c = \frac{1}{6\kappa^{1-\nu}}, \hat{c} = \frac{1}{54\kappa^{3-\nu}}$, where $\nu \in [0, 1]$

Recall Lemma C.2, Lemma C.4 and Lemma C.3, we have following inequalities:

$$\mathbb{E}\Phi(\bar{x}_{t+1}) \le \mathbb{E}\Phi(\bar{x}_t) - \left( \frac{\hat{c}\eta}{2} - \frac{\hat{c}^2 \eta^2 L}{2} \right) \mathbb{E}\|\bar{u}_{t+1}\|^2 + \frac{3\hat{c}\eta}{2} \mathbb{E}\|\bar{u}_{t+1} - \frac{1}{N} \sum_{i=1}^{N} \nabla_x f_i (x_t^i, y_t^i)\|^2$$

$$- \frac{\hat{c}\eta}{2} \mathbb{E}\|\nabla\Phi(\bar{x}_t)\|^2 + \frac{3\hat{c}\eta L_f^2}{N} \sum_{i=1}^{N} \mathbb{E}[\|x_t^i - \bar{x}_t\|^2 + \|y_t^i - \bar{y}_t\|^2] + \frac{3\hat{c}\eta L_f^2}{\mu} [\Phi(\bar{x}_t) - F(\bar{x}_t, \bar{y}_t)]$$

$$\mathbb{E}\|\bar{u}_{t+1} - \nabla_x \bar{F}_t\|^2 - \mathbb{E}\|\bar{u}_t - \nabla_x \bar{F}_{t-1}\|^2 = -\alpha \mathbb{E}\|\bar{u}_t - \nabla_x \bar{F}_{t-1}\|^2 + \frac{8L_f^2}{N^2 b}\sum_{i=1}^{N}\mathbb{E}[\hat{c}^2\eta^2\|u_t^i - \bar{u}_t\|^2$$

$$+c^2\eta^2\|v_t^i - \bar{v}_t\|^2] + \frac{4L_f^2}{Nb}\mathbb{E}[\hat{c}^2\eta^2\|\bar{u}_t\|^2 + c^2\eta^2\|\bar{v}_t\|^2] + \frac{2\alpha^2\sigma^2}{Nb}$$

$$\mathbb{E}\|\bar{v}_{t+1} - \nabla_y \bar{F}_t\|^2 - \mathbb{E}\|\bar{v}_t - \nabla_y \bar{F}_{t-1}\|^2 = -\beta \mathbb{E}\|\bar{v}_t - \nabla_y \bar{F}_{t-1}\|^2 + \frac{8L_f^2}{N^2 b}\sum_{i=1}^{N}\mathbb{E}[\hat{c}^2\eta^2\|u_t^i - \bar{u}_t\|^2$$

$$+ c^2\eta^2\|v_t^i - \bar{v}_t\|^2] + \frac{4L_f^2}{Nb}\mathbb{E}[\hat{c}^2\eta^2\|\bar{u}_t\|^2 + c^2\eta^2\|\bar{v}_t\|^2] + \frac{2\beta^2\sigma^2}{Nb}$$

$$[\Phi(\bar{x}_{t+1}) - F(\bar{x}_{t+1}, \bar{y}_{t+1})] - [\Phi(\bar{x}_t) - F(\bar{x}_t, \bar{y}_t)]$$

$$\leq -\frac{c\eta\mu}{2}[\Phi(\bar{x}_t) - F(\bar{x}_t, \bar{y}_t)] + \frac{\eta}{4\hat{c}}\|\hat{x}_{t+1} - \bar{x}_t\|^2 - \frac{\eta c}{4}\|\bar{v}_{t+1}\|^2 + \frac{3L_f^2 c\eta}{N}\sum_{i=1}^{N}\|x_t^i - \bar{x}_t\|^2$$

$$+\frac{3L_f^2 c\eta}{N}\sum_{i=1}^{N}\|y_t^i - \bar{y}_t\|^2 + 3c\eta\|\bar{v}_{t+1} - \nabla_y \bar{F}_t\|^2 \tag{48}$$

Next, we define a Lyapunov function, for any $t \geq 1$, we have $\Gamma_t = \Phi(\bar{x}_t) + \frac{3\hat{c}\eta}{2\alpha}\|\bar{u}_{t+1} - \frac{1}{N}\sum_{i=1}^{N}\nabla_x F(x_t^i, y_t^i)\|^2 + \frac{6\hat{c}L_f^2}{c\mu^2}[\Phi(\bar{x}_t) - F(\bar{x}_t, \bar{y}_t)] + \frac{18\hat{c}\eta L_f^2}{\mu^2\beta}\|\bar{v}_{t+1} - \nabla_y \bar{F}_t\|^2$.

$$\Gamma_{t+1} - \Gamma_t = -\left(\frac{\hat{c}\eta}{2} - \frac{\hat{c}^2\eta^2 L}{2}\right)\|\bar{u}_{t+1}\|^2 - \frac{\hat{c}\eta}{2}\|\nabla\Phi(\bar{x}_t)\|^2 + \frac{3\hat{c}\eta L_f^2}{N}\sum_{i=1}^{N}[\|x_t^i - \bar{x}_t\|^2 + \|y_t^i - \bar{y}_t\|^2]$$

$$+\frac{3\hat{c}\eta L_f^2}{\mu}[\Phi(\bar{x}_t) - F(\bar{x}_t, \bar{y}_t)] + \frac{3\hat{c}\eta}{2}\|\bar{u}_{t+1} - \frac{1}{N}\sum_{i=1}^{N}\nabla_x F(x_t^i, y_t^i)\|^2 - \frac{3\hat{c}\eta}{2}\|\bar{u}_{t+1} - \frac{1}{N}\sum_{i=1}^{N}\nabla_x F(x_t^i, y_t^i)\|^2$$

$$+\frac{12\hat{c}\eta L_f^2}{\alpha N^2 b}\sum_{i=1}^{N}\mathbb{E}[\hat{c}^2\eta^2\|u_{t+1}^i - \bar{u}_{t+1}\|^2 + c^2\eta^2\|v_{t+1}^i - \bar{v}_{t+1}\|^2] + \frac{6\hat{c}\eta L_f^2}{\alpha Nb}\mathbb{E}[\hat{c}^2\eta^2\|\bar{u}_{t+1}\|^2 + c^2\eta^2\|\bar{v}_{t+1}\|^2]$$

$$+\frac{3\hat{c}\eta\alpha\sigma^2}{Nb} - \frac{3\hat{c}\eta L_f^2}{\mu}[\Phi(\bar{x}_t) - F(\bar{x}_t, \bar{y}_t)] + \frac{3\hat{c}^2\eta L_f^2}{2c\mu^2}\|\bar{u}_{t+1}\|^2 - \frac{3\eta\hat{c}L_f^2}{2\mu^2}\|\bar{v}_{t+1}\|^2$$

$$+\frac{18L_f^4\hat{c}\eta}{N\mu^2}\sum_{i=1}^{N}\|x_t^i - \bar{x}_t\|^2 + \frac{18L_f^4\hat{c}\eta}{N\mu^2}\|y_t^i - \bar{y}_t\|^2 + \frac{18\hat{c}\eta L_f^2}{\mu^2}\|\bar{v}_{t+1} - \nabla_y \bar{F}_t\|^2$$

$$-\frac{18\hat{c}\eta L_f^2}{\mu^2}\mathbb{E}\|\bar{v}_{t+1} - \nabla_y \bar{F}_t\|^2 + \frac{144\hat{c}\eta L_f^4}{\beta N^2 b\mu^2}\sum_{i=1}^{N}\mathbb{E}[\hat{c}^2\eta^2\|u_{t+1}^i - \bar{u}_{t+1}\|^2 + c^2\eta^2\|v_{t+1}^i - \bar{v}_{t+1}\|^2]$$

$$+\frac{72\hat{c}\eta L_f^4}{Nb\beta\mu^2}\mathbb{E}[\hat{c}^2\eta^2\|\bar{u}_{t+1}\|^2 + c^2\eta^2\|\bar{v}_{t+1}\|^2] + \frac{36\beta\sigma^2\hat{c}\eta L_f^2}{Nb\mu^2} \tag{49}$$

Rearrange the terms, we have

$$\frac{\hat{c}\eta}{2}\|\nabla\Phi(\bar{x}_t)\|^2 = [\Gamma_t - \Gamma_{t+1}] - \left(\frac{\hat{c}\eta}{2} - \frac{\hat{c}^2\eta^2 L}{2} - \frac{3\hat{c}^2\eta L_f^2}{2c\mu^2} - \frac{6\hat{c}^3\eta^3 L_f^2}{\alpha Nb} - \frac{72\hat{c}^3\eta^3 L_f^4}{Nb\beta\mu^2}\right)\|\bar{u}_{t+1}\|^2$$

$$-\left(\frac{3\eta\hat{c}L_f^2}{2\mu^2} - \frac{6\hat{c}c^2\eta^3 L_f^2}{\alpha Nb} - \frac{72\hat{c}c^2\eta^3 L_f^4}{Nb\beta\mu^2}\right)\|\bar{v}_{t+1}\|^2 + \frac{36\beta\sigma^2\hat{c}\eta L_f^2}{Nb\mu^2} + \frac{3\hat{c}\eta\alpha\sigma^2}{Nb}$$

$$+\left(\frac{12\hat{c}\eta L_f^2}{\alpha N^2 b} + \frac{144\hat{c}\eta L_f^4}{\beta N^2 b\mu^2}\right)\sum_{i=1}^{N}\mathbb{E}[\hat{c}^2\eta^2\|u_{t+1}^i - \bar{u}_{t+1}\|^2 + c^2\eta^2\|v_{t+1}^i - \bar{v}_{t+1}\|^2]$$

$$+\left(\frac{18L_f^4\hat{c}\eta}{N\mu^2} + \frac{3\hat{c}\eta L_f^2}{N}\right)\sum_{i=1}^{N}[\|x_t^i - \bar{x}_t\|^2 + \|y_t^i - \bar{y}_t\|^2] \tag{50}$$

Taking the (50) telescoping sum over t from $s_t$ to $(s_t + 1)Q$, we have

$$\sum_{t=s_t}^{(s_t+1)Q-1} \|\nabla\Phi(\bar{x}_t)\|^2$$

$$\leq \sum_{t=s_t}^{(s_t+1)Q-1} \frac{2[\Gamma_t - \Gamma_{t+1}]}{\hat{c}\eta} - \left(1 - \hat{c}\eta L - \frac{3\hat{c}L_f^2}{c\mu^2} - \frac{12\hat{c}^2\eta^2 L_f^2}{\alpha Nb} - \frac{144\hat{c}^2\eta^2 L_f^4}{Nb\beta\mu^2}\right)\sum_{t=s_t}^{(s_t+1)Q-1}\|\bar{u}_{t+1}\|^2$$

$$- \left(\frac{3L_f^2}{\mu^2} - \frac{12c^2\eta^2 L_f^2}{\alpha Nb} - \frac{144c^2\eta^2 L_f^4}{Nb\beta\mu^2}\right)\sum_{t=s_t}^{(s_t+1)Q-1}\|\bar{v}_{t+1}\|^2 + \sum_{t=s_t}^{(s_t+1)Q-1}\left[\frac{72\beta\sigma^2 L_f^2}{Nb\mu^2} + \frac{6\alpha\sigma^2}{Nb}\right]$$

$$+\left(\frac{24L_f^2}{\alpha N^2 b} + \frac{288L_f^4}{\beta N^2 b\mu^2}\right)\sum_{t=s_t}^{(s_t+1)Q-1}\sum_{i=1}^{N}\mathbb{E}[\hat{c}^2\eta^2\|u_{t+1}^i - \bar{u}_{t+1}\|^2 + c^2\eta^2\|v_{t+1}^i - \bar{v}_{t+1}\|^2]$$

$$+\left(\frac{36L_f^4}{N\mu^2} + \frac{6L_f^2}{N}\right)\sum_{t=s_t}^{(s_t+1)Q-1}\sum_{i=1}^{N}[\|x_t^i - \bar{x}_t\|^2 + \|y_t^i - \bar{y}_t\|^2]$$

$$\leq \sum_{t=s_t}^{(s_t+1)Q-1} \frac{2[\Gamma_t - \Gamma_{t+1}]}{\hat{c}\eta} - \left(1 - \hat{c}\eta L - \frac{3\hat{c}L_f^2}{c\mu^2} - \frac{12\hat{c}^2\eta^2 L_f^2}{\alpha Nb} - \frac{144\hat{c}^2\eta^2 L_f^4}{Nb\beta\mu^2}\right)\sum_{t=s_t}^{(s_t+1)Q-1}\|\bar{u}_{t+1}\|^2$$

$$- \left(\frac{3L_f^2}{\mu^2} - \frac{12c^2\eta^2 L_f^2}{\alpha Nb} - \frac{144c^2\eta^2 L_f^4}{Nb\beta\mu^2}\right)\sum_{t=s_t}^{(s_t+1)Q-1}\|\bar{v}_{t+1}\|^2 + \sum_{t=s_t}^{(s_t+1)Q-1}\left[\frac{72\beta\sigma^2 L_f^2}{Nb\mu^2} + \frac{6\alpha\sigma^2}{Nb}\right]$$

$$+\left(\frac{24L_f^2}{\alpha N^2 b} + \frac{288L_f^4}{\beta N^2 b\mu^2}\right)\sum_{t=s_t}^{(s_t+1)Q-1}\sum_{i=1}^{N}\mathbb{E}[\hat{c}^2\eta^2\|u_{t+1}^i - \bar{u}_{t+1}\|^2 + c^2\eta^2\|v_{t+1}^i - \bar{v}_{t+1}\|^2]$$

$$+\frac{(\frac{36L_f^4}{\mu^2} + 6L_f^2)(Q-1)(q \times \frac{1}{20QL} \times \frac{1}{20QL})}{N}\sum_{t=s_t}^{(s_t+1)Q-1}\sum_{i=1}^{N}[[\hat{c}^2\sum_{i=1}^{N}\mathbb{E}\|u_{t+1,i} - \bar{u}_{t+1}\|^2 + c^2\mathbb{E}\|v_{t+1}^i - \bar{v}_{t+1}\|^2]]$$

where the last inequality holds dues to (40) and $\eta \leq \frac{1}{20QL}$ Then summing over all the restarts and divice by the T on the bothe size, we have

$$\frac{1}{T}\sum_{t=0}^{T-1}\mathbb{E}\|\nabla\Phi(\bar{x}_t)\|^2$$

$$\leq \frac{2[\Gamma_0 - \Gamma_T]}{\hat{c}\eta T} - \left(1 - \hat{c}\eta L - \frac{3\hat{c}L_f^2}{c\mu^2} - \frac{12L_f^2\eta^2\hat{c}^2}{Nb\alpha} - \frac{144\hat{c}^2\eta^2 L_f^4}{Nb\beta\mu^2}\right)\frac{1}{T}\sum_{t=0}^{T-1}\|\bar{u}_{t+1}\|^2$$

$$- \left(\frac{3L_f^2}{\mu^2} - \frac{12L_f^2\eta^2 c^2}{Nb\alpha} - \frac{144c^2\eta^2 L_f^4}{Nb\beta\mu^2}\right)\frac{1}{T}\sum_{t=0}^{T-1}\|\bar{v}_{t+1}\|^2 + \frac{6\alpha\sigma^2}{Nb} + \frac{72\beta\sigma^2 L_f^2}{Nb\mu^2}$$

$$+ \left(\frac{24L_f^2\eta^2}{Nb\alpha} + \frac{288L_f^4\eta^2}{Nb\beta\mu^2} + \frac{3 + 18\kappa^2}{200}\right)\frac{1}{NT}\sum_{t=1}^{T}\sum_{i=1}^{N}[[\hat{c}^2\sum_{i=1}^{N}\mathbb{E}\|u_{t+1}^i - \bar{u}_{t+1}\|^2 + c^2\mathbb{E}\|v_{t+1}^i - \bar{v}_{t+1}\|^2]]$$

$$\leq \frac{2[\Gamma_0 - \Gamma_T]}{\hat{c}\eta T} - \left(1 - \hat{c}\eta L - \frac{3\hat{c}L_f^2}{c\mu^2} - \frac{12L_f^2\eta^2\hat{c}^2}{Nb\alpha} - \frac{144\hat{c}^2\eta^2 L_f^4}{Nb\beta\mu^2} - \frac{\hat{c}^2\kappa^2}{12}\right)\frac{1}{T}\sum_{t=0}^{T-1}\|\bar{u}_{t+1}\|^2$$

$$- \left(\frac{3L_f^2}{\mu^2} - \frac{12L_f^2\eta^2 c^2}{Nb\alpha} - \frac{144c^2\eta^2 L_f^4}{Nb\beta\mu^2} - \frac{c^2\kappa^2}{12}\right)\frac{1}{T}\sum_{t=0}^{T-1}\|\bar{v}_{t+1}\|^2 + \frac{6\alpha\sigma^2}{Nb} + \frac{72\beta\sigma^2 L_f^2}{Nb\mu^2}$$

$$+ \left[\frac{\sigma^2(c_1^2 + c_2^2)}{30bL^2} + \frac{\zeta^2(c_1^2 + c_2^2)}{12L^2}\right]\kappa^2\eta^2 \tag{51}$$

where last inequality holds due to Lemma C.5 and the fact that $\alpha = c_1\eta^2, \beta = c_2\eta^2, c_1 = \frac{30L^2}{bN\kappa^{1-\nu}}, c_2 = \frac{30L^2}{bN\kappa^{2-2\nu}}, c = \frac{1}{6\kappa^{1-\nu}}, \hat{c} \leq \frac{1}{54\kappa^{3-\nu}}$, where $\nu \in [0,1], \kappa > 1$. Therefore, $\left(\frac{24L_f^2\eta^2}{Nb\alpha} + \frac{288\kappa^2 L_f^2\eta^2}{Nb\beta} + \frac{3+18\kappa^2}{200}\right)\max\{\hat{c}^2, c^2\} \leq \frac{24}{30\times36} + \frac{288\kappa^2}{30\times36} + \frac{3}{200\times36} + \frac{18\kappa^{2\nu}}{200\times36} \leq \frac{2}{5}\kappa^2$.

Furthermore, for the second term in the (51), $\eta \leq \frac{1}{20QL}$, then $1 - \hat{c}\eta L - \frac{3\hat{c}L_f^2}{c\mu^2} - \frac{12L_f^2\eta^2\hat{c}^2}{Nb\alpha} - \frac{144\hat{c}^2\eta^2 L_f^4}{Nb\beta\mu^2} - \frac{\hat{c}^2\kappa^2}{12} \geq 1 - \frac{1}{120} - \frac{3}{9} - \frac{12}{30} - \frac{144}{30\times36} - \frac{1}{12\times36} \geq 0$ where $\hat{c} \leq \frac{c}{9\kappa^2} = \frac{1}{54\kappa^{3-\nu}}$. In addition, for the third term in the (51), we have $\frac{3L_f^2}{\mu^2} - \frac{12L_f^2\eta^2 c^2}{Nb\alpha} - \frac{144c^2\eta^2 L_f^4}{Nb\beta\mu^2} - \frac{c^2\kappa^2}{12} \geq 3\kappa^2 - \frac{12}{30\times36} - \frac{144\kappa^2}{30\times36} - \frac{\kappa^{2\nu}}{12\times36} \geq 0$

$$\frac{1}{T}\sum_{t=0}^{T-1}\mathbb{E}\left\|\nabla\Phi\left(\bar{x}_t\right)\right\|^2$$

$$\leq \frac{2[\Phi(\bar{x}_0) - \Phi(\bar{x}_T)]}{\hat{c}\eta T} + \frac{3}{\alpha T}\mathbb{E}\|\bar{u}_1 - \nabla_x\bar{F}_0\|^2 + \frac{36L_f^2}{\mu^2\beta T}\mathbb{E}\|\bar{v}_1 - \nabla_y\bar{F}_0\|^2 + \frac{12L_f^2}{c\eta\mu^2 T}[\Phi(\bar{x}_0) - F(\bar{x}_0, \bar{y}_0)]$$

$$+ \frac{6\alpha\sigma^2}{Nb} + \frac{72\beta\sigma^2 L_f^2}{Nb\mu^2} + \left[\frac{\sigma^2(c_1^2 + c_2^2)}{30bL^2} + \frac{\zeta^2(c_1^2 + c_2^2)}{12L^2}\right]\kappa^2\eta^2$$

$$\leq \frac{2[\Phi(\bar{x}_0) - \Phi(\bar{x}_T)]}{\hat{c}\eta T} + \frac{3\sigma^2}{\alpha TBN} + \frac{36L_f^2\sigma^2}{\mu^2\beta TBN} + \frac{12L_f^2}{c\eta\mu^2 T}[\Phi(\bar{x}_0) - F(\bar{x}_0, \bar{y}_t)]$$

$$+ \frac{6\alpha\sigma^2}{Nb} + \frac{72\beta\sigma^2 L_f^2}{Nb\mu^2} + \left[\frac{\sigma^2(c_1^2 + c_2^2)}{30bL^2} + \frac{\zeta^2(c_1^2 + c_2^2)}{12L^2}\right]\kappa^2\eta^2 \tag{52}$$

Finally, we analyze the convergence of FedSGDA. $b = O(\kappa^\nu)$ for $\nu \in [0,1], c_1 = \frac{30L^2}{bN\kappa^{1-\nu}}, c_2 = \frac{30L^2}{bN\kappa^{2-2\nu}}, c = \frac{1}{6\kappa^{1-\nu}}, \hat{c} = \frac{1}{54\kappa^{3-\nu}}, T = \kappa^{3-\nu}T_0, Q = \frac{T_0^{1/3}}{N^{2/3}}, \eta = \frac{1}{20QL} = \frac{N^{2/3}}{20LT_0^{1/3}}$, we have $\alpha = c_1\eta^2 = \frac{3N^{1/3}}{40T_0^{2/3}b\kappa^{1-\nu}}, \beta = c_2\eta^2 = \frac{3N^{1/3}}{40T_0^{2/3}b\kappa^{2-2\nu}}. B = \frac{T_0^{1/3}b\kappa^{1-\nu}}{N^{2/3}}$

$$\frac{1}{T}\sum_{t=0}^{T-1}\mathbb{E}\left\|\nabla\Phi\left(\bar{x}_t\right)\right\|^2$$

$$\leq \frac{2160L[\Phi(\bar{x}_0) - \Phi^*]}{(NT_0)^{2/3}} + \frac{40\sigma^2}{\kappa^{3-\nu}(NT_0)^{2/3}} + \frac{480\sigma^2}{\kappa^2(NT_0)^{2/3}} + \frac{240L_f}{(NT_0)^{2/3}}[\Phi(\bar{x}_0) - F(\bar{x}_0, \bar{y}_0)]$$

$$+ \frac{9\sigma^2}{20b\kappa(NT_0)^{2/3}} + \frac{27\sigma^2}{5(NT_0)^{2/3}} + \left[\frac{3\sigma^2}{20b} + \frac{15\zeta^2}{40}\right]\frac{1}{(NT_0)^{2/3}}$$

$$\tag{53}$$

where $\Phi^*$ is the optimal. To let the right hand is less than $\varepsilon^2$, we get $T_0 = O(N^{-1}\varepsilon^{-3})$ and $T = O(\kappa^{3-\nu}N^{-1}\varepsilon^{-3})$. Considering the $b = \kappa^\nu$, Communication Complexity $\frac{T}{Q} = \kappa^{3-\nu}(NT_0)^{2/3} = \kappa^{3-\nu}\varepsilon^{-2}$. Sample complexity $bT = O(\kappa^3 N^{-1}\varepsilon^{-3})$. When $\nu = 1, b = \kappa$, Communication Complexity $\frac{T}{Q} = \kappa^2\varepsilon^{-2}$ $\qquad\square$

## D   Experiments

### D.1   Ablation Test results

In Fair classification, we also explore the combinations of the global and local learning rates of FedSGDA+ and report the test accuracy on CIFAR-10 in 3. Results show both global and local learning rates have a significant effect on the final performance.

In AUROC maximization, we present results of Fed-SGDA-M with different momentum parameters and local update number $Q$ in 2. When momentum parameters are 0.9 ( $1 - \alpha = 0.1$ and $1 - \beta = 0.1$), we put less weight on the history information and the FedSGDA-M tends to be local SGDA. When

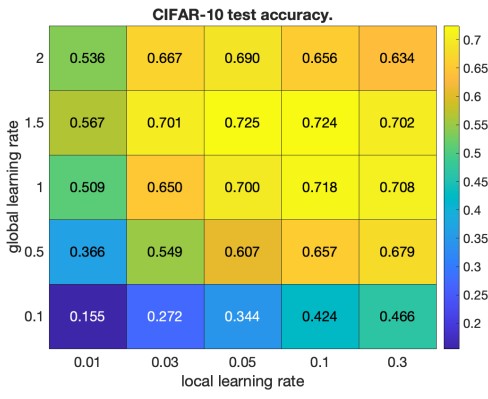

Figure 3: CIFAR-10 test accuracy on Fair Classification with various global learning rate and local learning rate $\eta_x$ ( $\eta_y = 0.1\,\eta_x$ )

the Q increase, the model performance decrease due to client drift. However, when the momentum parameter is 0.9, with the help of momentum variables, the effect of a larger local update number is reduced.

Table 2: CIFAR-10 test accuracy on AUROC maximization with various inner loop Q and momentum constant $\alpha$ / $\beta$

| momentum | 0.1 | 0.3 | 0.5 | 0.7 | 0.9 |
|---|---|---|---|---|---|
| Q = 10 | $0.6115 \pm 0.0055$ | $0.6029 \pm 0.0067$ | $0.6068 \pm 0.0068$ | $0.6027 \pm 0.0056$ | $0.6006 \pm 0.0070$ |
| Q = 20 | $0.6104 \pm 0.0040$ | $0.6059 \pm 0.0031$ | $0.6043 \pm 0.0068$ | $0.5988 \pm 0.0063$ | $0.5912 \pm 0.0085$ |
| Q = 50 | $0.6109 \pm 0.0069$ | $0.6043 \pm 0.0126$ | $0.6049 \pm 0.0171$ | $0.6027 \pm 0.0075$ | $0.5874 \pm 0.0093$ |

## D.2 Model Architectures

Table 3: Model Architecture for Fashion-MNIST

| Layer Type | Shape |
|---|---|
| Convolution + Tanh | $3 \times 3 \times 5$ |
| Max Pooling | $2 \times 2$ |
| Convolution + Tanh | $3 \times 3 \times 10$ |
| Max Pooling | $2 \times 2$ |
| Fully Connected + Tanh | 100 |
| Fully Connected + Tanh | 1 / 10 |

Table 4: Model Architecture for CIFAR-10

| Layer Type | Shape | padding |
|---|---|---|
| Convolution + ReLU | $3 \times 3 \times 16$ | 1 |
| Max Pooling | $2 \times 2$ | |
| Convolution + ReLU | $3 \times 3 \times 32$ | 1 |
| Max Pooling | $2 \times 2$ | |
| Convolution + ReLU | $3 \times 3 \times 64$ | 1 |
| Max Pooling | $2 \times 2$ | |
| Fully Connected + ReLU | 512 | |
| Fully Connected + ReLU | 64 | |
| Fully Connected + ReLU | 1 / 10 | |