# OpenReview forum: "Solving a Class of Non-Convex  Minimax Optimization in Federated Learning"
_NeurIPS.cc/2023/Conference — NeurIPS 2023 poster_

### Official Review · Reviewer_dmWe · 2023-06-28

**Soundness:** 3 good
**Presentation:** 2 fair
**Contribution:** 3 good
**Rating:** 6
**Confidence:** 3

**Summary:**

The paper focuses on the minimax optimization problem within the context of the federated learning framework. In order to address this problem, the authors introduced two algorithms named as FedSGDA+ and FedSGDA-M, to handle distinct types of losses, specifically focusing on the concavity of the maximizer component. The study presented a comprehensive analysis encompassing both theoretical and numerical evaluations.

**Strengths:**

This paper contributed two novel algorithms to solve various types of minimax optimization problems within the federated learning framework, which holds significant impact. As many machine learning problems are closely tied to minimax optimization, the developed algorithms offer valuable solutions for a wide range of applications. The solid theoretical analysis not only validates the effectiveness of their proposed algorithms but also lays a foundation for future investigations in the fields of minimax optimization and federated learning.

**Weaknesses:**

The paper needs presentation improvement as well as more clarification in the motivation and algorithm development.

Currently, I am inclining towards border acceptance with my questions below. I encourage authors to provide responses for my better assessment.

1. Line 27-29: The authors claim the federated learning is proposed to tackle the communication issue. But to my best knowledge, the federated learning has many advantages such as local data privacy preserving, utilizing more clients' data and enhancing computation power etc. Only highlighting communication advantages might be one-sided.

2. Line 145: The second inequality in (2) seems to be a strong assumption. It requires the deviation of the local data from the global average  to be bounded. However, under severe data heterogeneity and large parameters magnitude ,  such bound might not hold. Many federated learning works have studied to avoid such assumption, e.g. [1]. Can we also relax this assumption here?

3. Line 167: This assumption seems to be an addition assumption compared with the existing works in Table 1. Basically, it requires the gradient of $F$ w.r.t. $x$ to be bounded. It might not hold for many applications.

4. Wonder whether linear speed up w.r.t. to clients' number hold for this study. Can the authors provide some numerical validation?

5. For the algorithm presentation, can authors separate the server-side and client-side for better understanding?

6. Is FedSGDA-M motivated by the STORM-type of variance reduction method? Why is the algorithm not applicable for NC-C case?

7. Section 4.1: The loss looks be to a constrained one. How do the proposed algorithm hold the constraint?

8. Figure 2: Is FedSGDA in the figure the proposed FedSGDA+ or FedSGDA-M?

9. Can the authors provide numerical studies on the data heterogeneity, clients' number and computation cost?

[1] Karimireddy, Sai Praneeth, et al. "Scaffold: Stochastic controlled averaging for federated learning." International Conference on Machine Learning. PMLR, 2020.

**Questions:**

see weakness

**Limitations:**


The author acknowledged the need for further numerical studies and a more comprehensive analysis as a limitation of the paper. Since the research primarily focuses on theoretical algorithm development, there is no anticipated potential for negative societal impact.

---

> ### Author Rebuttal · Authors · 2023-08-03
>
> We are deeply thankful for your time and comments.
>
> > Q1. "The authors claim the federated learning is proposed to tackle the communication issue..."
>
> Thanks for your suggestion. Due to page limitation, I mention parts of advantages of FL, such as communication issue in Line 27 and privacy in Line 32. I will add more details about the advantages of FL.
>
> > Q2. "Line 145: The second inequality in (2) seems to be a strong assumption ... ". Many federated learning works have studied to avoid such assumption, e.g. [1] ...
>
> Variance Bound assumption is a popular assumption in federated optimization. [1] use
>
> > Q3. Line 167: This assumption seems to be an addition assumption compared with the existing works in Table 1.
>
> This assumption is used for the theoretical analysis in NC-C setting. We follow the only NC-C algorithm Local SGDA+ [2] in Table 1 and please see Assumption 6 in [2].
>
> > Q4. Wonder whether linear speed up w.r.t. to clients' number hold for this study.
>
> Remark 3.8 and Remark 3.14 show our two algorithms have linear speedup with respect to the number of worker nodes.
>
> > Q5. For the algorithm presentation, can authors separate the server-side and client-side for better understanding?
>
> Thanks for your suggestion. Because we only do average operation on the server, so we use denotes it as test mod (t, Q) = 0 in the Line 11 in algorithm 1 and Line 5 in the algorithm 2.
>
> > Q6. Is FedSGDA-M motivated by the STORM-type of variance reduction method? Why is the algorithm not applicable for NC-C case?
>
> FedSGDA-M uses a STORM-type variance reduction technique. The study of its application in NC-C case is in progress.
>
> > Q7. Section 4.1: The loss looks be to a constrained one. How do the proposed algorithm hold the constraint?
>
> We follow the task in [2] and constrained is not considered in the algorithm.
>
> > Q8. Figure 2: Is FedSGDA in the figure the proposed FedSGDA+ or FedSGDA-M?
>
> Thanks for the reminding. This is a typo. FedSGDA in the figure is FedSGDA-M.
>
>
>
> [1] Karimireddy, Sai Praneeth, et al. "Scaffold: Stochastic controlled averaging for federated learning." International Conference on Machine Learning. PMLR, 2020.
>
> [2] P. Sharma, R. Panda, G. Joshi, and P. Varshney. Federated minimax optimization: Improved convergence analyses and algorithms. In International Conference on Machine Learning, pages 19683–19730. PMLR,387 2022. https://proceedings.mlr.press/v162/sharma22c/sharma22c.pdf

---

> > ### Author Response · Authors · 2023-08-11
> > **Rebuttal**
> >
> > > Q2. "Line 145: The second inequality in (2) seems to be a strong assumption ... ". Many federated learning works have studied to avoid such assumption, e.g. [1] ...
> >
> > The Variance Bound assumption is a popular assumption in federated optimization and it is used in many existing FL minimax works [1] [2] and FL works [3] [4]. Thanks for your suggestion and the relaxation of this assumption will be considered in the future work.
> >
> > > Q5. "For the algorithm presentation, can authors separate the server-side and client-side for better understanding?"
> >
> > We will separate the algorithms in the final version to make it more clear.
> >
> >
> > > Q6. Is FedSGDA-M motivated by the STORM-type of variance reduction method? Why is the algorithm not applicable for NC-C case?
> >
> >
> > FedSGDA-M uses a STORM-type variance reduction technique. The current theoretical analysis of FedSGDA-M depends on the PL-condition assumption. The study of its application in the NC-C case is in progress.
> >
> >
> > > Q9. "Can the authors provide numerical studies on the data heterogeneity, clients' number and computation cost?"
> >
> > Since we cannot upload the images, I will add them to the final version.
> >
> >
> >
> > [1] Deng, Y. and Mahdavi, M., 2021, March. Local stochastic gradient descent ascent: Convergence analysis and communication efficiency. In International Conference on Artificial Intelligence and Statistics. PMLR. http://proceedings.mlr.press/v130/deng21a/deng21a.pdf
> >
> > [2] P. Sharma, R. Panda, G. Joshi, and P. Varshney. Federated minimax optimization: Improved convergence analyses and algorithms. In International Conference on Machine Learning, pages 19683–19730. PMLR,387 2022. https://proceedings.mlr.press/v162/sharma22c/sharma22c.pdf
> >
> > [3] Reddi, S., Charles, Z., Zaheer, M., Garrett, Z., Rush, K., Konečný, J., Kumar, S. and McMahan, H.B., 2020. Adaptive federated optimization. https://arxiv.org/pdf/2003.00295.pdf.
> >
> > [4] Khanduri, P., Sharma, P., Yang, H., Hong, M., Liu, J., Rajawat, K. and Varshney, P., 2021. Stem: A stochastic two-sided momentum algorithm achieving near-optimal sample and communication complexities for federated learning. Advances in Neural Information Processing Systems, 34, pp.6050-6061.

---

> > > ### Comment · Reviewer_dmWe · 2023-08-17
> > >
> > > I acknowedge that I have read the response by the authors. I raised my score from 5 to 6.

---

> > > > ### Author Response · Authors · 2023-08-17
> > > >
> > > > We are very grateful for your positive comments.

---

### Official Review · Reviewer_Nj11 · 2023-07-01

**Soundness:** 3 good
**Presentation:** 2 fair
**Contribution:** 2 fair
**Rating:** 6
**Confidence:** 2

**Summary:**

this paper proposed new algorithms for solving federated minimax problems in both nonconvex concave and nonconvex PL (or strongly concave) settings under the assumption of data heterogeneity. the authors propose a novel way to incorporate the variance reduction technique into the federated setting and provide solid convergence analysis in both settings. numerical experiments on auc maximization and fairness problems are reported to show the advantage of the proposed algorithms.

**Strengths:**

1. the proposed algorithms achieve the best convergence rate compared with existing works in the same setting. the final rate comes with linear speedup under the heterogeneous setting.
2. the idea of updating y with a fixed x in the concave case helps to improve communication complexity.
3. the numerical experiments are convincing. the authors compare with the majority of existing works.
4. the paper is easy to follow.

**Weaknesses:**

the step size \eta needs to be very small to guarantee the convergence since it's inverse propositional to q, the local updates. this requirement hurts the performance in practice

**Questions:**

both algorithms in this paper use the same batch data to compute the x gradient and the y gradient. in that case, those gradients (\nabla_x f and \nabla_y f) are not independent of each other. will this cause problems in the proof? can the author double-check?

**Limitations:**

the algorithm names in the table are not consistent with the names in the algorithms.

---

> ### Author Rebuttal · Authors · 2023-08-02
>
> Thanks a lot for your time and comments.
>
> > Q1. "the step size \eta needs to be very small to guarantee the convergence since it's inverse propositional to q, ...."
>
> Thanks for your mention. The relationship between learning rate $\eta$ and local training iterations q is easy to understand because if we do not put a restriction on the learning rate, a large q will lead to divergence due to client drift in federated learning. In addition, this relationship is common in federated learning. Such as Corollary1 in [1] $\eta_l = \Theta (1/KL \sqrt{T})$ where K is the local update and Theorem 1 in [2] shows that $\eta_y \leq \frac{1}{8L_f \tau}, \frac{\eta_x}{\eta_y} \leq \frac{1}{8\kappa^2} $ where $\tau$ is the local training iterations. The relationship between the local step size and the local training iterations guarantees convergence.
>
> > Q2. "... those gradients (\nabla_x f and \nabla_y f) are not independent of each other ..."
>
> This has brought significant challenges to the proof and thus some works use double-loop architecture, namely multiple samples and update for y but only one step for x.  We use simpler single-loop architecture. We use \nabla_x f and \nabla_y f to update x and y, separately. Lemma B.1, Lemma B.3 and Lemma C.2, Lemma C.3 show how to balance variable x and y.  This challenge also shows the value of our work.
>
>
> [1] Reddi, S., Charles, Z., Zaheer, M., Garrett, Z., Rush, K., Konečný, J., Kumar, S. and McMahan, H.B., 2020. Adaptive federated optimization. https://arxiv.org/pdf/2003.00295.pdf.
>
> [2] Sharma, P., Panda, R., Joshi, G. and Varshney, P., 2022, June. Federated minimax optimization: Improved convergence analyses and algorithms. In International Conference on Machine Learning (pp. 19683-19730). PMLR. https://proceedings.mlr.press/v162/sharma22c/sharma22c.pdf

---

### Official Review · Reviewer_sra2 · 2023-07-06

**Soundness:** 2 fair
**Presentation:** 3 good
**Contribution:** 2 fair
**Rating:** 3
**Confidence:** 4

**Summary:**

(1) This paper describes two algorithms, FedSGDA+ and FedSGDA-M, for solving non-convex minimax optimization problems.
(2) Theoretical guarantees are establised for these two algorithms, under several assumptions.
(3) Empirical tests are conducted.

**Strengths:**

**Originality:**

The techniques used in this paper are not novel from an optimization perspective. However, they can indeed be applied to minimax problems and have a theoretical bound.

**Quality:**

The notations, problem statements, and mathematical proofs appear rigorous, although I have not scrutinized them word by word.

**Clarity:**

The readability of this article is good.

**Significance:**

This paper provides agorithms with better convergence rates (although some extra assumptions were made in the paper, ). The techniques (and tricks) may inspire and guide future researchers.

**Weaknesses:**

1. The algorithms described in "Federated minimax optimization: Improved convergence analyses and algorithms" [36] and the ones in this paper are very similar. Specifically, FedSGDA-M and Local SGDA-M in [36] are basically the same, and the difference between FedSGDA+ and Local SGDA in [36] is very small, only in the step size, which may not be the most crucial part of the FedSGDA+ algorithm. Therefore, I believe that there is not much improvement in the algorithm aspect of the paper.

2. On the other hand, the convergence rates established in this paper appear to be better than previous algorithms, but the assumptions used are different from those of the previous. Thus, this comparison seems somewhat unfair and does not necessarily indicate a substantial improvement in convergence rates.

3. As for the experiments, firstly, this paper does not describe whether the experimental settings conform to the theoretical assumptions, so it is unclear whether the empirical studies can support the theoretical results. Secondly, the relationship between the complexity established in this paper and the number of worker nodes is not reflected in the experiments. Lastly, only Local SGDA was put into the experimental baseline set, and some algorithms that perform well in this field, such as FEDNEST and SAGDA, unfortunately did not become the experimental baselines.


**Questions:**

I have detailed the issues I identified as weaknesses in the previous message. Please address these questions I posed to them.

**Limitations:**

Yes, they have.

---

> ### Author Rebuttal · Authors · 2023-08-02
>
> Thank you for your recognition of the quality, clarity, and significance of our paper.
>
> >Q1. "... FedSGDA-M and Local SGDA-M in [1] are basically the same, and the difference between FedSGDA+ and Local SGDA in [1] is very small, ..."
>
> I respectively disagree with this statement. SGDA is a very classical algorithm in minimax optimization and many minimax algorithms are designed based on it. 1) Although both FedSGDA-M and Local SGDA-M are momentum-based algorithms, they are completely different since our FedSGDA-M introduces a momentum-based variance reduction technique, and reduces communication complexity and sample complexity. 2) FedSGDA+ and Local SGDA+ are both used for NC-C settings but the introduction of global learning in FedSGDA+  obviously improves the communication complexity. It should be mentioned that Local SGDA+  in [1] is from [2] but it provides a better theoretical analysis. Since [1] does not come up with a new algorithm for NC-C setting,  the convergence rate of Local SGDA+ cannot match our results.
>
> >Q2. "... the assumptions used are different from those of the previous. ..."
>
> The assumption for FedSGDA+ and Local SGDA+ are $\textbf{completely same}$. Although Assumption Assumption 3.11. is different from the one used in [1], it is still a widely used assumption in optimization analysis. Many typical centralized stochastic algorithms use this assumption, such as SREDA [3], and VR-SMDA [4]. Similarly, it is also used in FL algorithms such as MIME [5] and Stem [6].
>
> >Q3. "... experimental settings  ..."
>
> We consider two tasks. 1) Fair Classification is from [1] and since FedSGDA+ and Local SGDA+ use the same assumptions, we compare these two methods in NC-C settings. 2) AUPRC maximization is from [7]. Since [7] not only uses a similar assumption (Assumption 1 (iv)), but uses a stronger assumption (Assumption 1 (ii)), we use this task to evaluate our algorithms.
>
> > Q4. "... the relationship between the complexity established in this paper and the number of worker nodes is not reflected in the experiments ..."
>
> I provide the theoretical analysis in our works. Since we cannot upload the images in rebuttal, we will add it in final version.
>
> > Q5  "... only Local SGDA was put into the experimental baseline set... "
>
> I completely disagree with the statements. I politely remind you that Local SGDA and Local SGDA+ in [1] are two different algorithms and only Local SGDA+ is used for NC-C setting [1]. SAGDA [8] considers the NC-PL setting and cannot be used for NC-C setting. This is why we only use Local SGDA+ as a baseline in Fair Classification. In the AUROC Maximization tasks, we use local SGDA, CODA+, Momentum SGDA, CODASCA, and SAGDA as baselines, not only Local SGDA.
>
> [1] P. Sharma, R. Panda, G. Joshi, and P. Varshney. Federated minimax optimization: Improved convergence analyses and algorithms. In International Conference on Machine Learning, pages 19683–19730. PMLR,387 2022
>
> [2] Y. Deng and M. Mahdavi. Local stochastic gradient descent ascent: Convergence analysis and communication efficiency. In International Conference on Artificial Intelligence and Statistics, pages 1387–1395. PMLR, 2021.
>
> [3] Luo, L., Ye, H., Huang, Z. and Zhang, T., 2020. Stochastic recursive gradient descent ascent for stochastic nonconvex-strongly-concave minimax problems. Advances in Neural Information Processing Systems, 33.
>
> [4] Feihu Huang, Xidong Wu, and Heng Huang. Efficient mirror descent ascent methods for nonsmooth minimax problems. Advances in Neural Information Processing Systems, 34, 2021.
>
> [5] Karimireddy, S.P., Jaggi, M., Kale, S., Mohri, M., Reddi, S.J., Stich, S.U. and Suresh, A.T., 2020. Mime: Mimicking centralized stochastic algorithms in federated learning. arXiv preprint arXiv:2008.03606.
>
> [6] Khanduri, P., Sharma, P., Yang, H., Hong, M., Liu, J., Rajawat, K. and Varshney, P., 2021. Stem: A stochastic two-sided momentum algorithm achieving near-optimal sample and communication complexities for federated learning. Advances in Neural Information Processing Systems, 34, pp.6050-6061.
>
> [7] Zhishuai Guo, Mingrui Liu, Zhuoning Yuan, Li Shen, Wei Liu, Tianbao Yang. Communication-Efficient Distributed Stochastic AUC Maximization with Deep Neural Networks. ICML 2020
>
> [8] Yang, Haibo, Zhuqing Liu, Xin Zhang, and Jia Liu. "SAGDA: Achieving
>  Communication Complexity in Federated Min-Max Learning." arXiv preprint arXiv:2210.00611 (2022). https://openreview.net/pdf?id=wTp4KgVIJ5

---

> > ### Comment · Reviewer_sra2 · 2023-08-17
> > **Thank you for your response**
> >
> > Thank you for your response. I will consider your feedback.

---

> > > ### Author Response · Authors · 2023-08-17
> > >
> > > Thanks. Looking forward to your reply.

---

> ### Author Response · Authors · 2023-08-14
> **Thanks for your review**
>
> We truly thank your review. Since the discussion period already began, we will appreciate it if you can check our responses and let us know if there are any further questions.

---

> ### Author Response · Authors · 2023-08-17
> **Thanks for your review**
>
> I wanted to express our gratitude for taking the time to review our work and for providing feedback. We have submitted a rebuttal addressing the concerns and comments raised in your review.
> We kindly request that you take a moment to review our rebuttal because the discussion will end soon. Your feedback is of utmost importance to us, as it will help ensure the quality and rigor of our work. Thanks.

---

### Official Review · Reviewer_7re5 · 2023-07-07

**Soundness:** 3 good
**Presentation:** 3 good
**Contribution:** 3 good
**Rating:** 7
**Confidence:** 4

**Summary:**

This paper studied a class of federated nonconvex minimax optimization problems. Authors proposed FL algorithms and provided sample complexity under three different settings including nonconvex-concave, nonconvex-strongly conconcave, and nonconvex-PL conditioned. Authors showed that derived rates has the best sample complexity. Experimental results demonstrated the superiority.

**Strengths:**

1. The theoretical analysis is one main contribution of this paper. It's nontrivial to derive sample complexity for different settings.
2. Experiments also validated the efficacy of proposed algorithms.
3. Code is available and looks neat.

**Weaknesses:**

I have a major concern regarding the discussion and comparison of existing works in the paper. Specifically, when addressing local algorithms, the authors solely focus on SGDA (with and without momentum), which is considered a classical and relatively old algorithm. However, it is worth noting that several recent works have demonstrated improvements in convergence rates. For instance, in [1], Section 4.2 provides insights into enhanced rates, while [2] presents Theorems C.9 that showcase accelerated convergence rates for convex-nonconcave minimax problems. Additionally, [3] introduces Theorem 4.1, which is relevant to this discussion. Although these works may not directly address federated learning, their findings on accelerated convergence rates for general minimax problems should be brought into the discussion to avoid confusion among readers in the future.

### reference:
1. Mahdavinia et.al., Tight Analysis of Extra-gradient and Optimistic Gradient Methods For Nonconvex Minimax Problems
2. He et.al., GDA-AM: ON THE EFFECTIVENESS OF SOLVING MIN-IMAX OPTIMIZATION VIA ANDERSON MIXING, ICLR 2022
3  Lee et.al., Fast Extra Gradient Methods for Smooth Structured Nonconvex-Nonconcave Minimax Problems, Neurips 2021


**Questions:**

See above

**Limitations:**

Authors discussed limitations, although generic.

---

> ### Author Rebuttal · Authors · 2023-08-02
>
> Thanks a lot for your valuable and constructive comments!
>
> >Q1. " ... existing works in the paper ... Although these works may not directly address federated learning, their findings on accelerated convergence rates for general minimax problems "
>
> Thanks a lot for your suggestions. Due to page limitations, we pay more attention to FL minimax works. I added these minimax works in Sec 2.1 Single-Machine Minimax. Given that we cannot resubmit the new version, we will present them in the final version.

---

> > ### Comment · Reviewer_7re5 · 2023-08-14
> > **Rebuttal received**
> >
> > I've read the rebuttal and other reviewers' comments. Most concerns are addressed. I have one additional question:
> >
> > ### Q1: the use of GDA.
> > GDA has **poor convergence properties** for solving general minimax problems. Simultaneous GDA tends to diverge. Even alternating GDA can only show bounded convergence. This phenomenon is hidden when solving deep learning problems and applications considered in the paper. But it is problematic when solving bilinear/quadratic games or general 1d minimax functions. Have authors tried replacing GDA with our algorithms and what's the performance?

---

> > > ### Author Response · Authors · 2023-08-14
> > > **Thanks for your response**
> > >
> > > We sincerely thank the response from the reviewer.
> > >
> > > Minimax training is usually difficult and as shown in [1], the choices of the learning rate for x and y are important. We show the relationship between two learning rates in Corollary 3.7. and Corollary 3.13. In addition, in deep learning problems, the selection of learning rate of variable x, i.e. model parameter updating, is more critical.
> > >
> > > In this paper, we mainly focus on GDA (Gradient Descent Ascent) based approaches as GDA is still the most commonplace algorithm to solve minimax problems, especially when solving deep learning problems and applications considered in this paper. We note [2] conducts an in-depth investigation of limitations of GDA algorithm (e.g., smaller learning rate, cycling/divergence issue) and gives a systematic analysis of how to improve GDA dynamics. Nonetheless, integration of the proposed GDA improvement is not within this paper’s scope. We will discuss the limitations pointed out by [2] in greater detail in our final version. And we thank the reviewer for pointing out a future direction.
> > >
> > >
> > > [1] Lin, Tianyi, Chi Jin, and Michael Jordan. "On gradient descent ascent for nonconvex-concave minimax problems." In International Conference on Machine Learning, pp. 6083-6093. PMLR, 2020.
> > >
> > > [2] Huan He, Shifan Zhao, Yuanzhe Xi, Joyce Ho, Yousef Saad, GDA-AM: ON THE EFFECTIVENESS OF SOLVING MIN-IMAX OPTIMIZATION VIA ANDERSON MIXING, ICLR 2022

---

> > > > ### Comment · Reviewer_7re5 · 2023-08-17
> > > >
> > > > I am glad the relationship between two learning rates is discussed because twotime scale updating rule is necessary to avoid non local minmax points. I asked it just for curiosity. I have elevated my score as all my concerns have been addressed, and I am glad to see its publication.

---

> > > > > ### Author Response · Authors · 2023-08-17
> > > > >
> > > > > Thank you very much for your recognition of our work!

---

### Official Review · Reviewer_uFXg · 2023-07-12

**Soundness:** 4 excellent
**Presentation:** 3 good
**Contribution:** 4 excellent
**Rating:** 7
**Confidence:** 4

**Summary:**

Authors propose federated stochastic gradient descent-ascent method for solving minimax problems in federated learning setting and demonstrate that oracle and communication complexity of their method is significantly better than that of analogues in nonconvex-(concave/non-concave/PL) cases with respect to dependence on desired accuracy.

**Strengths:**

Theoretical analysis of the proposed method shows that its complexity is the best among alternatives, construction of the algorithm is simple enough, important practical cases of nonconvex and PL functions are considered.

**Weaknesses:**

I guess, within the goals of the paper there are no significant weaknesses: theoretical results are good and well-justified, minimal experiments confirm the advantages of the proposed method.

**Questions:**

I suggest authors to reflect deviation of accuracy for runs with different realisations of randomness on their figures in addition to particular realisation of convergence curve — for example, with shadow. Its important for understanding stability of advantage of the proposed method. And maybe the title should be changed to correctly reflect the content of the paper: "solving a class" seems to have unclear meaning, etc.

**Limitations:**

Everything is okay.

---

> ### Author Rebuttal · Authors · 2023-08-02
>
> Thanks a lot for your appreciation. I will add extra deviation of accuracy in the final version and modify the paper according to your suggestion. Thanks!

---

> > ### Comment · Reviewer_uFXg · 2023-08-19
> >
> > Dear authors, thank you for your work on the final version of your paper! The rebuttal has clarified my questions. I decided to keep my overall rating the same.

---

### Decision · Program_Chairs · 2023-09-21

**Decision:**

Accept (poster)

**Comment:**

Dear Authors,

Thank you for submitting your paper on non-convex minimax optimization in federated learning. The paper received some positive feedback; the reviewers appreciated the simple construction of your algorithm, the theoretical analysis, and the experiments that confirm the efficiency of your method.

I would also like to point out that your paper contains many weaknesses. Luckily, during the rebuttal phase, you addressed most of the concerns from the reviewers, but please make sure to incorporate them into the final version.

Best,

AC